# Development of Rhombus Hanging Basket Walking Track Robot for Cantilever Casting Construction in Bridges

**Yuping Ouyang** [1,2,*], **Jiarui Huang** [1,2] **and Kaifang Song** [1,2]

1   School of Mechanical and Electrical and Vehicle Engineering, East China Jiaotong University, Nanchang 330013, China; 18146713048@163.com (J.H.); songkaifang1997@163.com (K.S.)
2   National and Local Joint Engineering Research Center for Fruit Intelligent Photoelectric Detection Technology and Equipment, Nanchang 330013, China
*   Correspondence: ouyuping1987@163.com

**Abstract:** For enhancing the efficiency of cantilever casting construction in a bridge, a novel rhombus traveling track for a hanging basket was developed and optimized. The business-oriented software called MIDAS was applied to analyze the mechanical properties of the hanging basket with a full load and a control load. The strength and distortion of the walking mechanism are within the specified range when the maximum beam unit internal force is 149.69 MPa, which is less than the allowable stress of steel of 175 Mpa, and the anti-overturning safety factor of the hanging basket is 2.5, which meets the requirements. Through the comparative analysis of the key components and the finite element calculation, it was found that there is about a 30% redundancy in the structure performance. Therefore, further optimization of each structure was carried out, and the front elevation of the main truss of 20° was achieved. It obtained the best performance in all aspects.

**Keywords:** cantilever construction; rhombus hanging basket; walking track; finite element simulation; mechanical characteristics

## 1. Introduction

The cantilever pouring construction method of bridges is one of the most important methods to solve the construction problems of the river-crossing bridge. The rhombic hanging basket has a stable structure, high utilization efficiency, and convenient disassembly and movement, so it is the most widely used method in cantilever construction [1,2]; the method generally adopts the hanging basket as the construction technology. The hanging basket can bear the load and beam weight in the construction process. It is designed without a unified, systematic and perfect standard. Hence, before the application of the hanging basket, a reasonable design and accurate mechanical checking calculation can be carried out to ensure its safety in the construction process [3,4].

In the meantime, domestic and foreign scholars have carried out relevant research [5,6]. Li, 2018 used a prestressed concrete continuous beam bridge as a support during the construction of the Mao-ji-te Bridge and introduced the differences due to the difference in the load-bearing structure. Four types of hanging baskets are listed. The main structure and force characteristics of different structural types of hanging baskets were analyzed in detail [7]. Gao, 2016 shortened the length of the hanging basket by setting a triangular truss on the cast-in-place main beam. He combined the advantages of the front fulcrum hanging basket and the middle fulcrum hanging basket and proposed a short platform composite traction rope hanging basket to improve the problem that the main beam section can be changed when the long platform traction rope hanging basket is walked [8]; the above studies involve the introduction of different hanging basket structures during cantilever beam pouring and summarized the matters needing attention to ensure the safe construction of the hanging basket. However, there is still a lack of finite element modeling analysis and discussion on the hanging basket's construction and walking. Chen et al.,

2013 described the key points of the main truss of the triangular hanging basket in the design process and carried out spatial modeling, analysis, and calculation of the structure through the MIDAS/Civil program, which put forward theoretical support for the finite element simulation of the cantilever hanging basket construction in the future [9]; An et al., 2018 proposed that the high pier long-span continuous rigid frame aqueduct in Xujiawan be constructed by the cantilever method, and that the first-order method of ANSYS be used for optimization calculation [10]. The original rhombic hanging basket is transformed. Apart from these, a cable-stayed combined light rhombic hanging basket combining the traditional rhombic hanging basket and the sliding cable-stayed hanging basket has been proposed, which reduced the quality of the hanging basket and improved the utilization coefficient; Gu et al., 2019 added a transportation system and hoisting system to the traditional hanging basket structure. After the corrugated steel web was lifted to the poured beam surface, the transportation system moved it to the hanging basket hoisting starting point [11]. He et al., 2020 introduced a new asynchronous pouring construction technology. It made full use of the excellent shear resistance of corrugated steel webs to support the hanging basket, optimized the walking system, increased the construction platform, effectively accelerated the construction speed, reduced the construction load, and saved on project cost [12]; Zhou et al., 2020 combined the excellent shearing capacity of the corrugated steel mesh and introduced asynchronous pouring and rapid construction methods. A significant change is that they used the corrugated steel mesh itself as the main load-bearing member to support the hanging basket and beam sections. In the new method, the balance state of the new hanging basket system is optimized from the cantilever to the pure support state, and the walking system is optimized in the meantime, which greatly shortened the construction time and ensured the safety of the project [13].

Generally speaking, the research so far has mainly focused on the optimization and transformation of the main truss structure of the cradle. The shortcomings of the walking system of the cradle still exist, and there are few studies in this area. Relevant industrial standards have not yet been established and it is difficult to set them up in hanging basket-related equipment. Consequently, it is indispensable to calculate and simulate the mechanical properties of different equipment in order to ensure the safety of the construction process. Relying on the Tianjin Hai-he Bridge project, this paper proposed a new type of traveling rail for the cradle and optimized the structure of the main truss of the rhombus cradle, which solved the problem of low efficiency in the construction of the traditional cradle. Through calculation and finite element simulation, the hanging basket met the engineering requirements.

## 2. Materials and Methods

This section introduces the project overview of the Tianjin Hai-he River Bridge. In view of the shortcomings of the past bridge, in the construction process, a basket walking track with a simple walking mode and lighter weight was designed to be used in the Hai-he River Bridge project. The track is made of a new type of polymer. The manufacturing process of the track and materials is introduced, and the track is modeled. The mechanical properties of the track materials were tested according to the Chinese national standards. The structure of the hanging basket is introduced and modeled, and different working conditions in the construction are described. In order to verify the reliability of the rails and cradles made of new polymer materials used in practical projects, according to the models of the cradles and tracks established in the previous model, Abaqus and business-oriented software MIDAS/Civil were used to simulate and analyze the rails and cradles, analyze the stability of the rails and cradles, and make the structures met the safety requirements in the project.

### 2.1. Project Profile

The Hai-he Bridge is an extra-large bridge in the southern section of the Tang-Jin Expressway Expansion Project. It is designed to cross the Tian-jin Hai-he River, and the

total length is 2850 m. The main bridge is a three-span continuous girder structure with a span combination of (100 + 160 + 100) m. Shown in Figure 1 is the longitudinal section. The height of the main pier is 19.5 m, and the upper part of the main bridge adopted a prestressed concrete continuous box girder. The section is a single box and single cell with straight webs. Shown in Figure 2 is the cross section of the suspended cast block. The beam height of the fulcrum of the main span, the beam height of the middle span, the top width of the single box beam, the bottom plate width, and the box beam flange width are 9.5 m, 3.5 m, 13.25 m, 6.25 m and 3.5 m. And the beam height changes according to a 1.8 times parabola. There are 18 suspended cast blocks in total, of which block 9 is the heaviest, with a concrete volume of 79.3 m$^3$, a weight of 206.96 t and a length of 4.5 m.

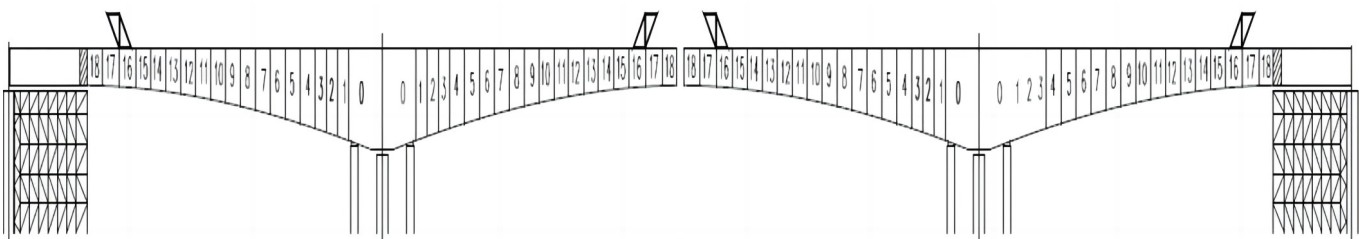

**Figure 1.** The longitudinal section of the main bridge.

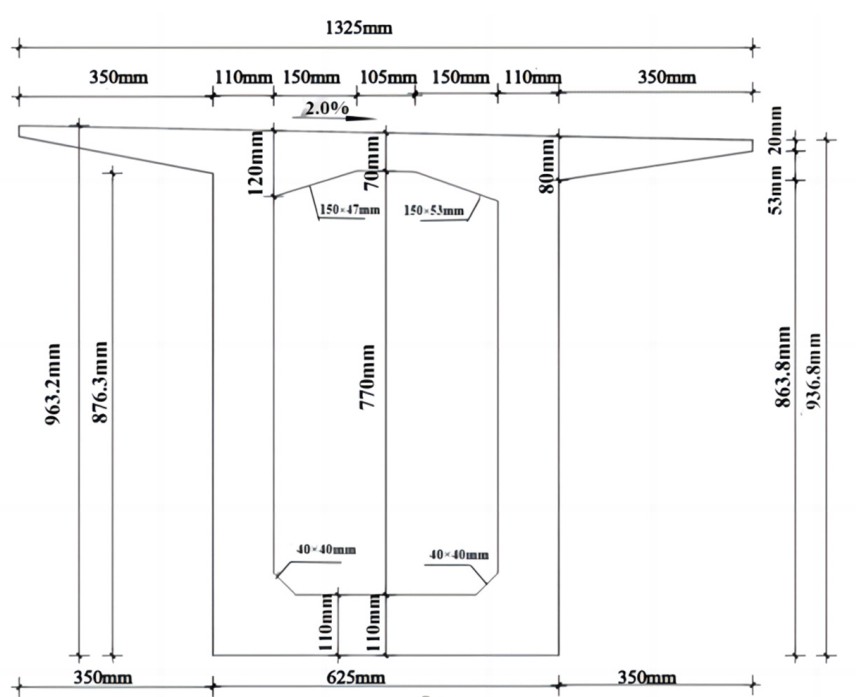

**Figure 2.** Cross section of suspended pouring block.

*2.2. Structure of Rhombus Traveling Track of Hanging Basket*

2.2.1. Structure of Walking Track

A    Introduction to the Structure of the Walking Track

The track height and center of gravity of the traditional hanging basket walking mechanism are high, which increases the risk of the hanging basket overturning [14]. Simultaneously, the walking track is long and bulky, and the laying and dismantling process is complex, which limits the improvement of engineering efficiency [15]. Based on the above problems, a new type of walking track of the hanging basket was designed, which is light and easy to be disassembled and assembled. Its single-chip chain plate structure is shown in Figure 3.

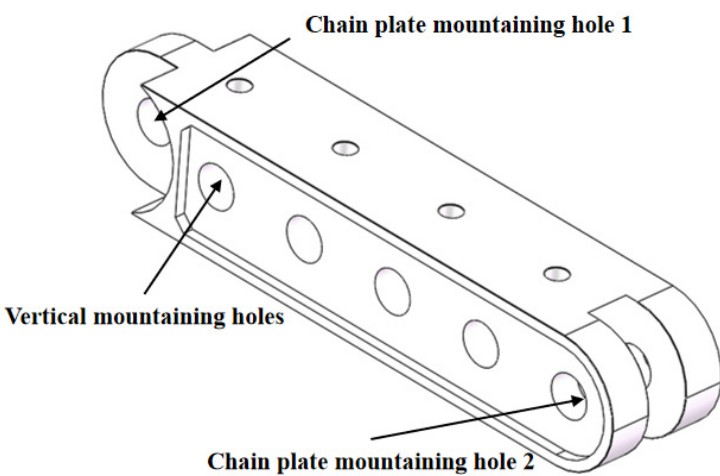

**Figure 3.** New type of walking track chain plate structure.

B　　Preparation of Walking Track

　　　The material preparation equipment for the new type of hanging baskets' walking track is a twin-screw extruder, as shown on the left side of Figure 4. The preparation method is to add 3% of SWR-3F nylon toughening agent, 2% of 2-Hydroxybenzophenone, 3% of organic phosphate and 50% of PA6 chips in the mixing bucket shown in Figure 4 according to the mass ratio. After mixing, uniformly, put it into the extruder through the hopper, heat it, and feed the material into the middle of the extruder. According to the mass ratio, add 15% of LEE powder, 5% of KH550 coupling agent and 22% of glass fiber. And vacuumize, heat all the materials in the extruder to 260 °C, and enter the cooling pool to cool after molding, then shear and granulate it into fine particles, as shown on the right side of Figure 4. The glass fiber and ultrafine powder particles treated with the coupling agent are added to improve the tensile strength of the material [16,17], and SWR-3F toughening agent is added to improve the impact resistance of the material [18–20].

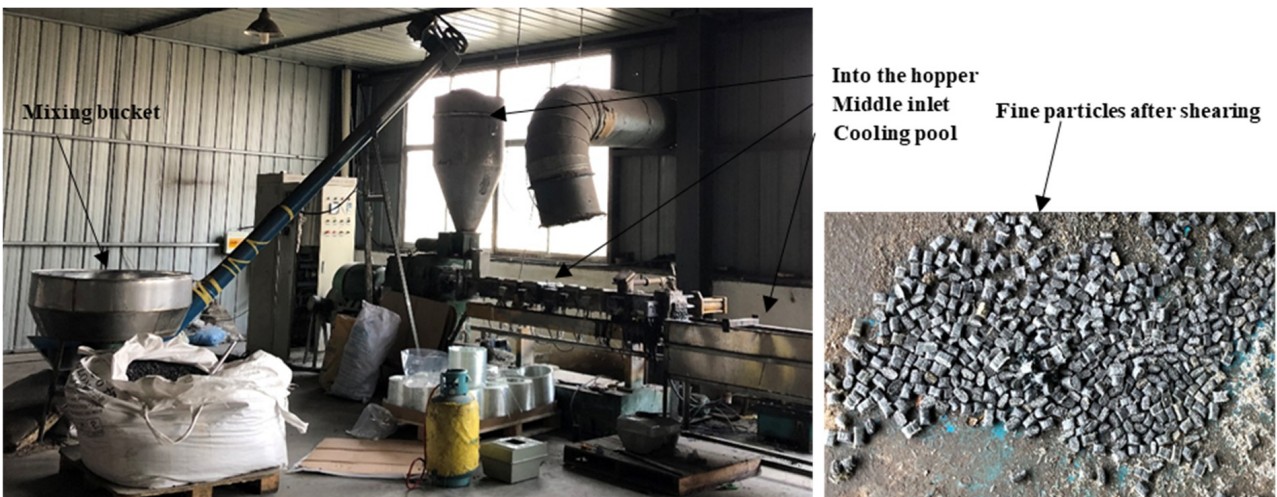

**Figure 4.** Twin screw extruder and granulation.

　　　When the new hanging basket track is used, the convex part of the first chain plate is inserted into the recessed part of the second chain plate, so that the chain plate mounting hole 1 of the first chain plate can be aligned with the chain plate mounting hole 2 of the second chain plate. Bolts are inserted into the aligned holes for connection. In the same measure, multiple chain plates can be connected to form a load-bearing chain plate track. For the baskets of different sizes, several rows of the load-bearing chain plate tracks can be arranged side by side, so that the vertical mounting holes on the side-by-side chain

plates are aligned and inserted with bolts on connection. Therefore, the bearing area of chain-bearing plate tracks can be increased. In this project, each rhombic truss has four load-bearing chain plates, as shown in Figure 5. The gap between the middle load-bearing chain plate tracks served to avoid contacting the finishing steel bar on the bridge deck. The front support of the main truss of the hanging basket is installed on the walking trolley. When the hanging basket is walking, the hydraulic push rod applies force on the connecting pin fixed on the chain plate. Due to the counter-effect of force, the hydraulic push cylinder travels away from the axis pin to complete the walking of the hanging basket equipment.

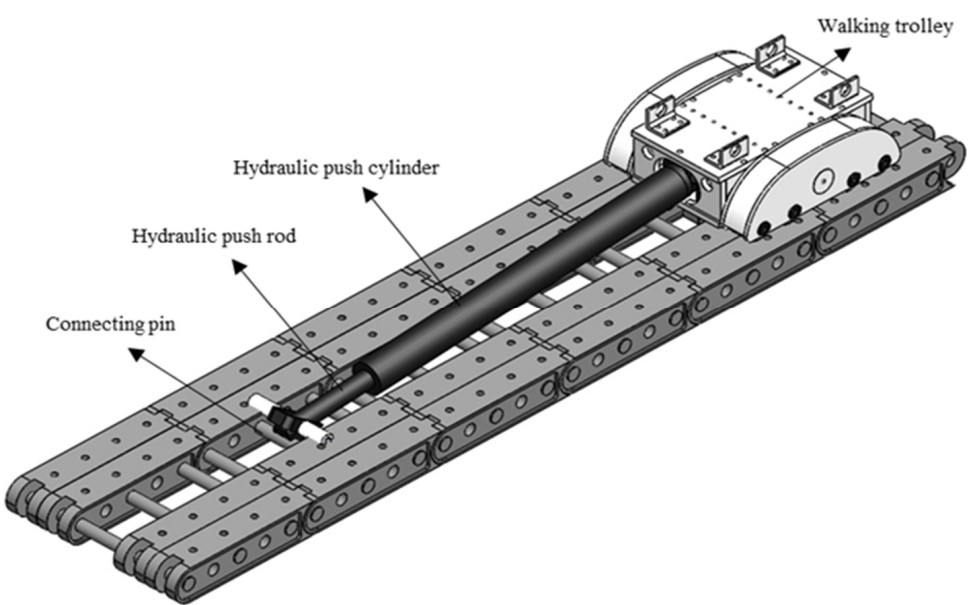

**Figure 5.** Load bearing chain plate track.

C    Material Performance Test of Walking Track

The new walking track material is composed of 3% SWR-3F nylon toughener, 2% 2-hydroxyphenyl benzophenone, 3% phosphate ester, 50% nylon 6 chips, 15% LEE powder, 5% KH550 coupling agent and 22% glass fiber. Because of the new materials used in the track, this study tested the mechanical properties according to the Chinese national standard (GB2918) [21]. The standard requires that the test sample should be in a constant environment and the test temperature should be 18 °C–28 °C, and this is suitable for the mechanical property test of plastic samples. According to the actual stress condition and working environment of the hanging basket, the tensile performance, compressive performance, and impact resistance of the new materials are selected to be tested.

1.    Tensile Property Test

The tensile property test schedule is shown in Table 1. Firstly, the material, which is easy to fix by the fixture, is produced according to the standard, as shown in Figure 6a. The width and thickness of the tensile part in the middle of the sample are 10 mm and 4 mm and the tensile section area is 40 mm$^2$, as shown in Figure 6b.

**Table 1.** Tensile property test schedule.

| | |
|---|---|
| Experimental instrument | Electronic universal testing machine |
| Experimental temperature | 20 °C |
| Stretching speed | 50 mm/min |
| Test repetition times | 5 |
| Number of test samples | 5 |

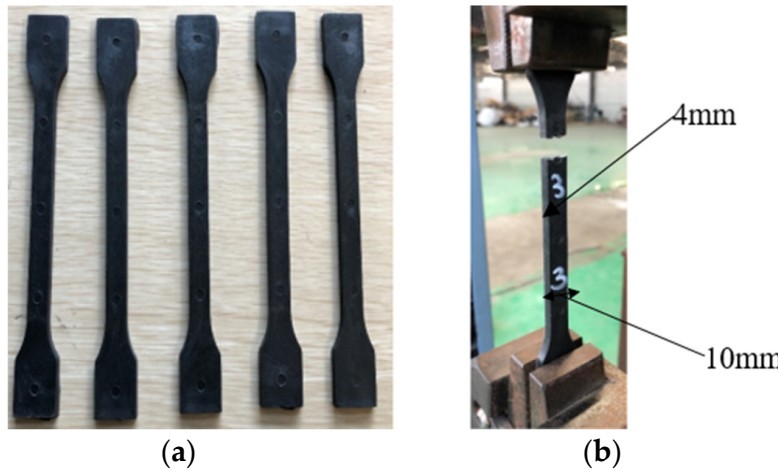

**Figure 6.** New materials in tensile test: (**a**) before the test, (**b**) after the test.

The device used in the experiment is an electronic universal testing machine. The test sample was in a constant environment. The test environment temperature was 20 °C. The test method is as follows: Firstly, the materials tested are clamped tight according to the set gauge length, and the distance between the upper clamp and the lower one is 120 mm, as shown in Figure 6b. Secondly, calibration and zero-setting are conducted in the software, and the tensile test is started at the speed of 50 mm per min. The experiment is terminated at the moment when the material is fractured. The experiment shown in Figure 6 was repeated five times with five samples. The five groups of experimental results were summarized and averaged, as shown in Figure 7. Before point A, under the condition that the sample was not completely tightened by the clamp, the test force was relatively low and would not be enhanced during the process of trying to clamp. The sample was totally tightened by the clamp at the time of 1.30 s (point A) when the test force started to increase. At the time of 9.9 s (point B), the test force and stress reached the maximum, followed by a small and temporary drop; then the material was broken and the testing machine stopped loading to finish the experiment. The stress at point B served as the maximum stress of the sample, which is the tensile strength of the material, with a measured value of 95.55 MPa.

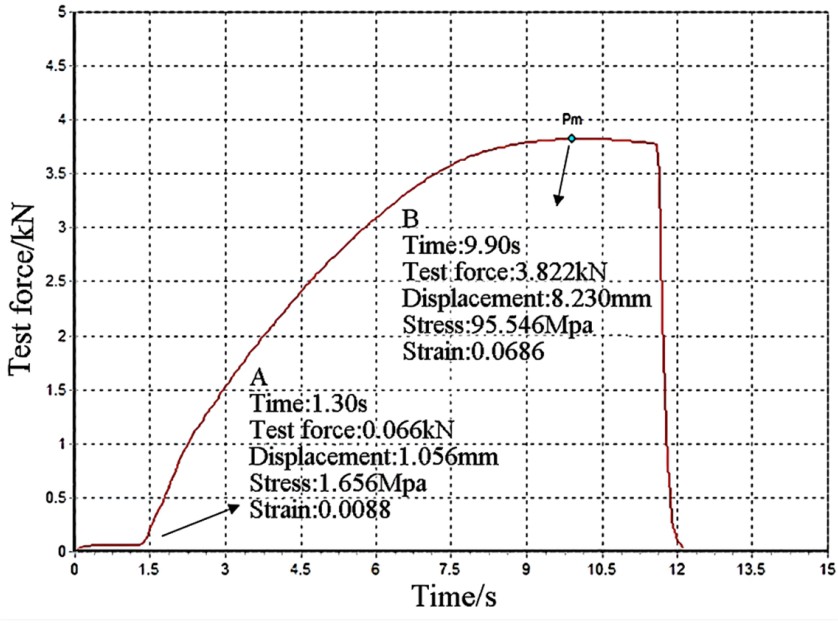

**Figure 7.** Time and test force curve of tensile test sample.

2. Compressive Performance Test

The tensile property test schedule is shown in Table 2. The compressive strength of the new materials needed to be tested by testing their compressive properties. The sample used for the test is shown in Figure 8. The compressive strength test applied a uniform load in the direction of side A. The length and width of the sample is 10.5 mm, the compression area is 110.25 mm$^2$ and the height is 30 mm.

**Table 2.** Schedule of compression performance test.

| | |
|---|---|
| Experimental instrument | Electronic universal testing machine |
| Experimental temperature | 20 °C |
| Compression speed | 10 mm/min |
| Test repetition times | 5 |
| Number of test samples | 5 |

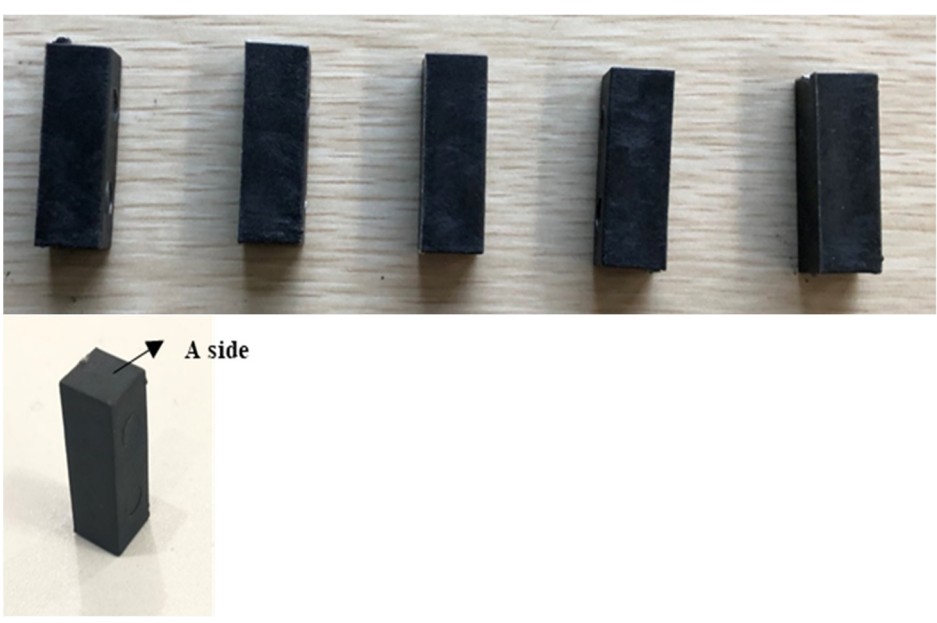

**Figure 8.** Compression test specimens.

The instrument used in the test is an electronic universal testing machine. The test sample was in a constant environment. The test environment temperature was 20 °C. The test method is as follows: Firstly, the remaining material at both ends of the sample is polished by sandpaper, and the A side of the sample should be placed upward on the stage. Calibration and zeroing are then performed in the computer software, and the compression test is started at 10 mm/min and stopped when the sample collapses. The experiment shown in Figure 8 was repeated five times with five samples. The test was repeated for 5 groups, and the samples that collapsed due to compression are shown in Figure 9. The parameters and results of each group of tests needed to be recorded. The five groups of experimental results were summarized and averaged, as shown in Figure 10. Before point A, the sample was not entirely compacted by the testing machine, and the stress caused by the slight unevenness of the sample surface after compaction was relatively small. From point A, the sample was under complete stress, which was increased sharply. At the beginning of point B, the increase of the stress then witnessed a downturn and reached its maximum at point C. After point C, the material was fractured upon the maximum stress, resulting in a drastic drop in the stress. Then, the testing machine stopped loading, and the experiment came to an end. The stress at point C is the tensile strength of the sampled material with a measured value of 109.88 MPa.

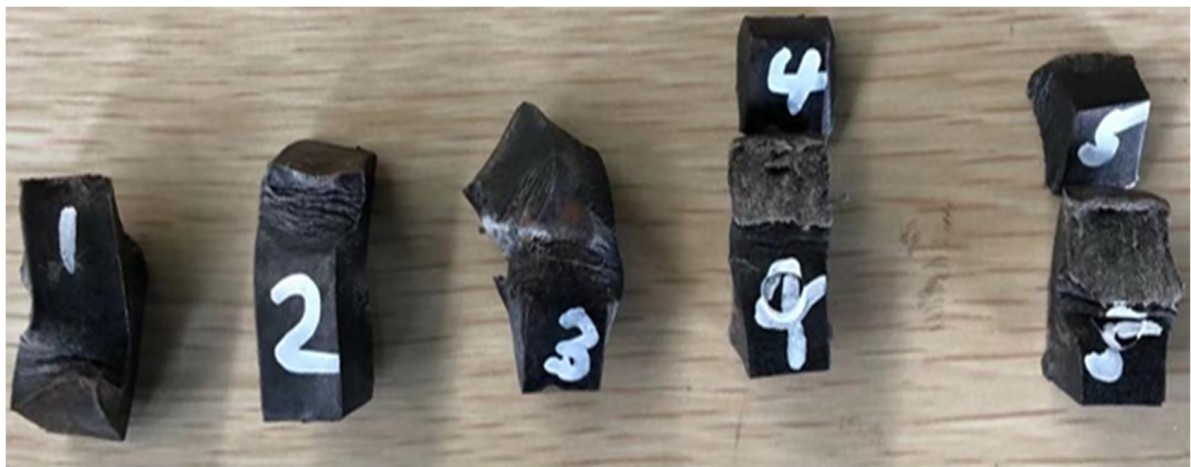

**Figure 9.** Specimens collapsed in the compression test.

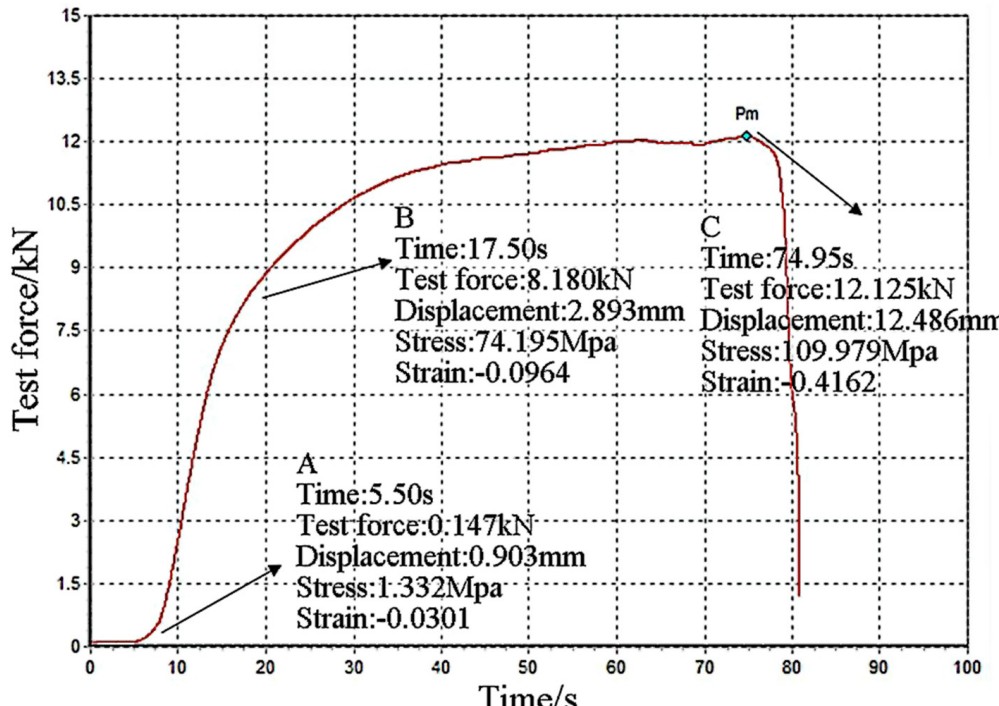

**Figure 10.** Time and force curve of compression test.

3. Impact Resistance Test

The tensile property test schedule is shown in Table 3. Standard size material is produced according to the standard. The length of the sample, the width and thickness are 80 mm, 10 mm and 4 mm. And the cross-sectional area is 40 mm², as shown in Figure 11.

**Table 3.** Impact resistance test schedule.

| | |
|---|---|
| Experimental instrument | Impact testing machine |
| Experimental temperature | 20 °C |
| Pendulum energy | 7.5 J |
| Impact speed | 3.8 m/s |
| Test repetition times | 5 |
| Number of test samples | 5 |

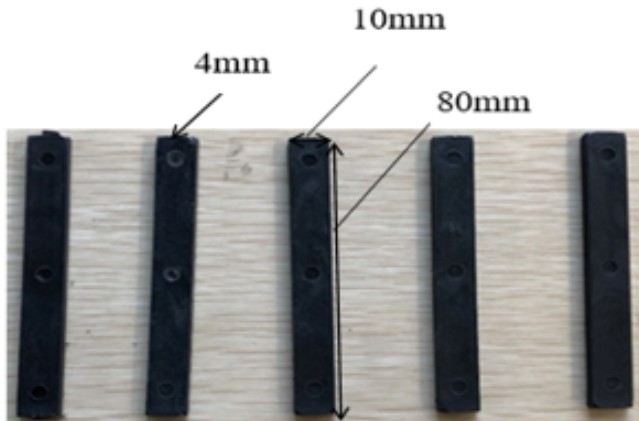

**Figure 11.** New material in impact resistance test.

The instrument used in the test is an electronic simple suspension combined impact testing machine. Samples were made according to the standard, and the number of measurements was of 5 groups. The test sample was in a constant environment. The test environment temperature was 20 °C, and the test method was as follows: The pendulum is fixed to the upper part of the instrument. The sample is placed on the impact table. With the bounce button pressed, the sample is broken by the falling pendulum bob, and the corresponding data are recorded by the device. The broken samples after the experiment are shown in Figure 12. The results were summarized upon the completion of the present five groups of experiments. As shown in the table, the "angle of elevation" refers to the angle between the originally fixed position of the pendulum and the vertical line; the "air attack angle" refers to the calculated angle at which the pendulum rotated in the air after deducting the thickness of the material; the "impact angle" refers to the angle between the highest point reached on the other side after the pendulum broke the sample and the vertical line. It is analyzed that the smaller the impact angle is, the larger the impact strength of the material is. The calculation of the impact strength $a_{cU}$ of the specimen is shown in Equation (1), where $E_{cU}$ is the energy absorbed by the measured pattern. The experiment sample shown in Figure 11 was repeated five times. The five groups of experimental results were summarized and averaged, as shown in Table 4. It can be seen from Table 4 that the impact angle of test group 4 is larger because the material is not uniform in the molding process, resulting in a thicker position in the middle and more material, so the impact strength is increased slightly. The impact strength of the material is 67.65 kJ/m$^2$.

$$a_{cU} = \frac{E_{cU}}{h_{cU}b_{cU}} \times 10^3 \tag{1}$$

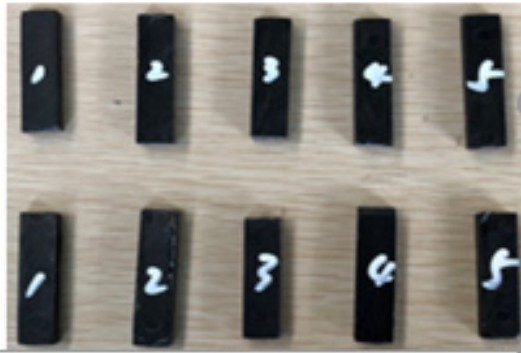

**Figure 12.** Sample broken after test.

**Table 4.** Summary of impact test results.

| Experiment | Elevation [°] | Air Attack Angle [°] | Impact Angle [°] | Absorbed Energy [J] | Impact Strength [kJ/m²] |
|---|---|---|---|---|---|
| 1 | 160 | 158.40 | 101.92 | 2.797 | 69.92 |
| 2 | 160 | 158.40 | 103.63 | 2.685 | 67.12 |
| 3 | 160 | 158.40 | 107.23 | 2.452 | 61.30 |
| 4 | 160 | 158.40 | 98.55 | 3.022 | 75.55 |
| 5 | 160 | 158.40 | 105.36 | 2.575 | 64.37 |

### 2.2.2. Structure of Hanging Basket

#### A    Introduction to the Structure of the Hanging Basket

A cantilever casting rhombic hanging basket was adopted in constructing the main span of this project. The basket is mainly composed of 2 rhombic main trusses and connecting parts, which constitute the main load-bearing system, bottom basket, suspension system, rear anchor, walking system and formwork system. The transverse distance between the two rhombic main trusses is 5.7 m. The upper and lower chords of the main truss consist of $400 \times 300 \times 12$ mm rectangular steel pipe, while the vertical web member is made of $350 \times 250 \times 10$ mm rectangular steel pipe; the front tilt web member under compression is composed of $400 \times 300 \times 12$ mm rectangular steel tubes, the rear tilt web member under compression consists of $420 \times 300 \times 12$ mm grooved steel, and the casing pipe guide beam of the lower chord is made of $350 \times 250 \times 12$ mm rectangular steel pipe. The upper beam is composed of a double-split $200 \times 400 \times 10$ mm rectangular steel pipe, the front and rear beams of the bottom basket consist of $400 \times 300 \times 12$ mm rectangular steel pipe, and the longitudinal beam of the bottom basket is made of HN $150 \times 300 \times 6.5 \times 9$–6000 mm H-type steel. The hanger consists of double $\Phi 32$ fin twisted steel bar. The transverse door union is made of a square steel tubular truss with a vertical height of 1400 mm and a transverse width of 500 mm. The chord consists of $120 \times 120 \times 5$ mm steel pipe, and the web member is composed of $100 \times 50 \times 4$ mm steel pipe; the outer guide beam is made of $350 \times 250 \times 12$ mm rectangular steel tube, and the main structural steel materials of the hanging basket are Q235 steel. The structure diagram of the main truss is shown in Figure 13.

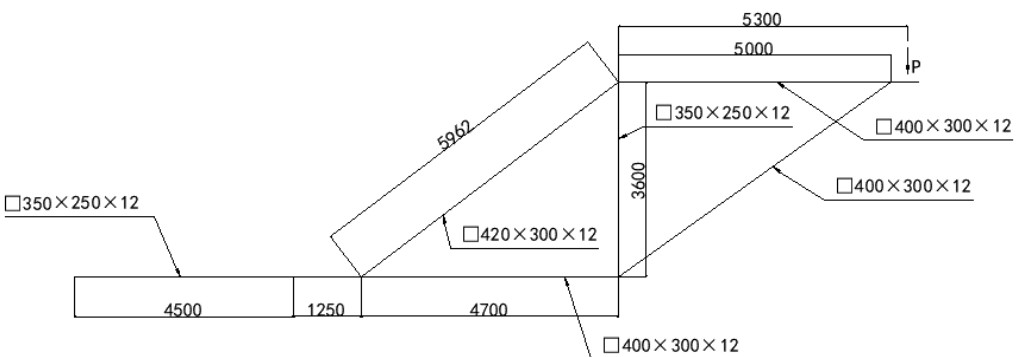

**Figure 13.** Dimensions of the main truss.

The weight of the hanging basket is 63 t. In the worst case, the total weight of the formwork is 22.8 t. The static load coefficient and dynamic load coefficient of the hanging basket are 1.2 and 1.4, respectively. In the running process of the hanging basket, the two most typical working conditions are selected for analysis, both of whose wind speed is 13.6 m/s vertically downward. Among them, working condition 1 refers to the construction process of segment 9 (with the heaviest pouring concrete). Working condition 2 refers to the hanging basket of the walking state.

B    The Construction of Finite Element Model of the Hanging Basket

The MIDAS/Civil analysis applied in this paper is especially suitable for the simulation of bridge structure. MIDAS/Civil software is a simulation and analysis software based on finite element theory developed by POSCO Group in Korea in the last century. It has the following advantages over ANSYS, Abaqus, and other general finite element simulation software.

In the pre-processing stage, the software has its own unit model that meets all kinds of standards, according to the actual situation of the project, such as the type of cross section of the structure, the material, and the force units, such as truss unit, beam unit, plate unit, tensile and compression unit, etc. In the post-processing stage, if the structure is subjected to a variety of loads, then the combination of the loads of the different analyses can be generated according to the design specifications automatically, which saves a lot of time for the operator.

Firstly, a three-dimensional model was established in SolidWorks 2020 SP0.0 software according to the entity of the cradle. According to the established three-dimensional diagram, the model was imported into MIDAS/Civil 2022 v1.2 software to establish the finite element simulation model of the cradle, as shown in Figure 14.

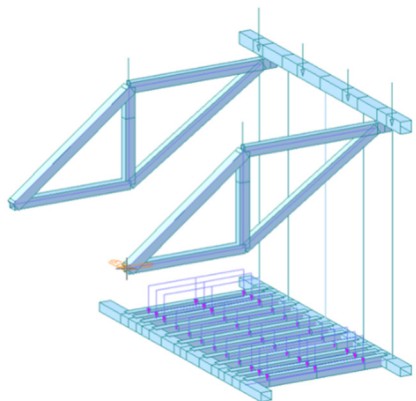

**Figure 14.** Overall calculation model of hanging basket.

C    Working Condition of the Hanging Basket

The wind direction of the three conditions was vertical downward (wind speed 13.6 m/s). Condition 1 and condition 2 were calculated by MIDAS/Civil, and condition 3 was calculated by referring to the Chinese national standard. In condition 1, the hanging basket was located at the 9# section, which is the heaviest section of 4 m, and the weight of pouring concrete was 125t. When the concrete pouring was completed but not solidified, the stress and distortion of the hanging basket in the working state were calculated; in condition 2, the hanging basket walked to the end of the newly poured section, and did not mount the concrete itself. In condition 3, the hanging basket was in the walking state, and the stability of the hanging basket in the working state was calculated.

*2.3. Simulation of the Walking Track and the Hanging Basket*

2.3.1. Finite Element Simulation of New Material for Walking Track

According to Chinese national standards, the above experimental results of the new material of the cradle track demonstrated that the structure has high tensile strength, high compressive strength and good impact resistance. In the meantime, finite element simulations have been carried out to verify its reliability in practical engineering applications. In order to verify the reliability of the hanging basket track chain plate made of new polymer materials in practical engineering, the model of a single track chain plate would be established in SolidWorks software.

The track of the hanging basket was paved on the surface of the bridge, on which the main truss carrying the to-be-cast concrete was walking and moving. Firstly, the

boundary conditions for the simulation were set to constrain the freedom degree of the bottom surface of the model (without other structural support). The track chain plates were installed embedded into each other, so there were some missing material positions under the surface of a single chain plate, as shown at point A of Figure 15. Therefore, for convenient calculation, the bearing area of the hanging basket track was reduced without calculating the place where the material was missing, and the evenly distributed load was applied to the effective areas. The calculation was done under the condition of the maximum bearing capacity of 8 tracks operating. In addition, the safety coefficient and the evenly distributed load stood at 1.1 and 142.39 t, respectively, as shown in Figure 16.

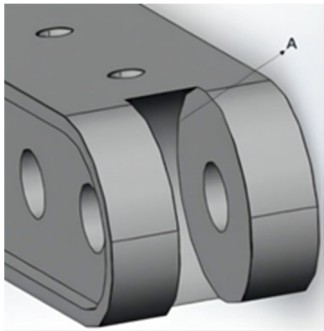

**Figure 15.** Missing points of internal material.

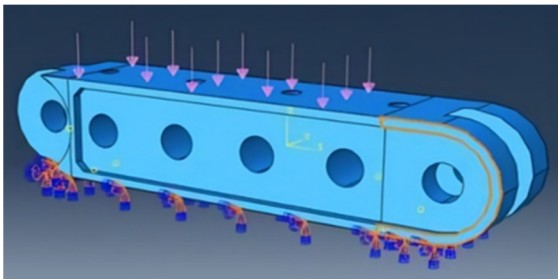

**Figure 16.** Boundary conditions and uniform load of chain plate.

From the simulation results shown in Figure 17, the position of the overall deformation distribution of the track where it changed the most is on both sides of the surface of the hole position. The maximum deformation is 0.087 mm. Thus, the simulation results proved that the track chain plate made of the new material can be applied in the construction process of the hanging basket.

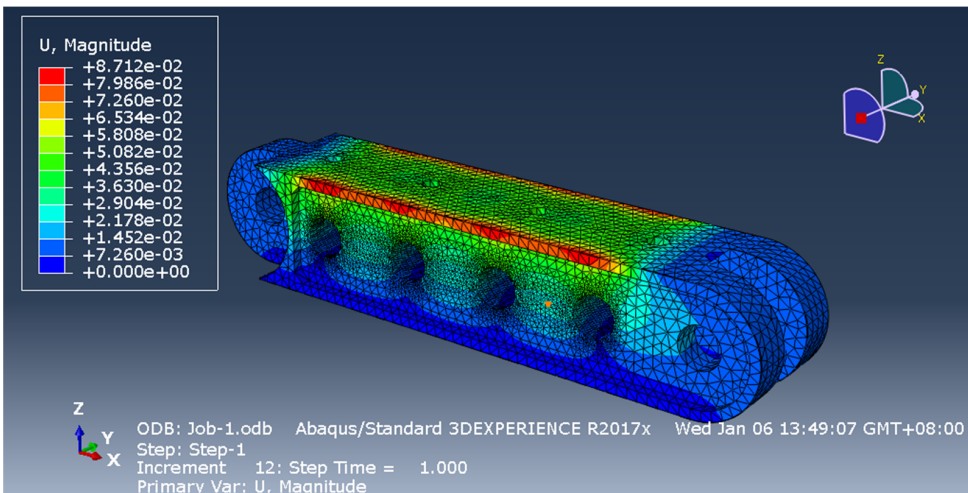

**Figure 17.** Mechanical simulation results.

2.3.2. Finite Element Simulation of the Hanging Basket Structure Based on MIDAS/Civil

The load was applied to calculate the stress and distortion of the hanging basket under the condition of the No. 9 section of the construction (the heaviest pouring concrete) in condition 1. The overall calculation model of the hanging basket is shown in Figure 14. In the cantilever casting of the construction, the hanging basket equipment was generally divided into three states, as discussed in Section 2.2.2 ((C) above). The combined load analysis was adopted.

A    Analysis of Mechanical Performance of the Hanging Basket under Condition 1

We applied the above finite element model, applied the load and analyzed and calculated the deformation and stress of the structure. In Figures 18 and 19, the calculated stress cloud diagram and vertical distortion cloud diagram of the hanging basket structure can be seen, respectively. As can be seen from Figure 18, the vertical distortion of 12.53 mm occurred at the upper end of the main truss, which is less than 1/400 of the length of the member (13.25 mm); the vertical displacement of 25.00 mm occurred in the middle of the front upper beam, which is 12.47 mm, which is less than 1/400 of the length of the member (30 mm) due to the reference system. The vertical displacement of 28.65 mm occurred in the middle of the front lower beam, which is 16.12 mm. The distortion of the structure met the design requirements.

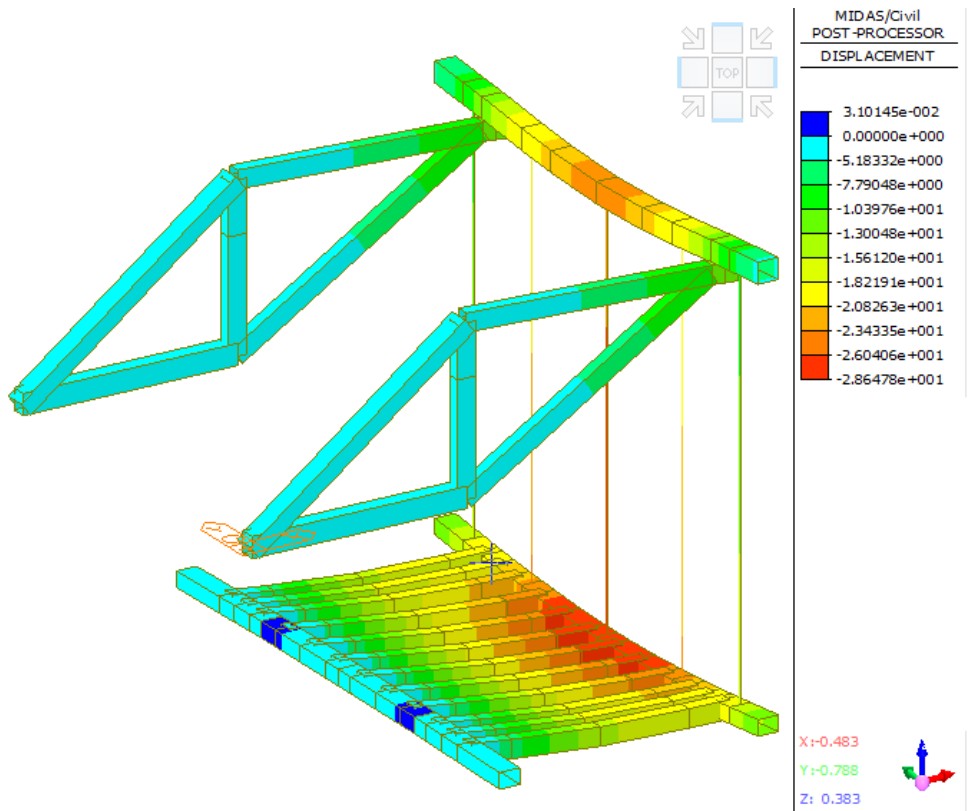

**Figure 18.** Z-direction distortion nephogram of the whole structure of the hanging basket under condition 1.

As can be seen from Figure 19, the maximum beam unit internal force generated by the whole structure is 149.69 MPa, which occurred at the junction of the front upper beam and the main truss. The stress of the structure is less than the allowable stress of Q235 steel $\left[\sigma_{Q235}^{\alpha}\right] = 175\text{MPa}$ [22], which met the design requirements of the project. In addition, it is indispensable to analyze the main components of the hanging basket.

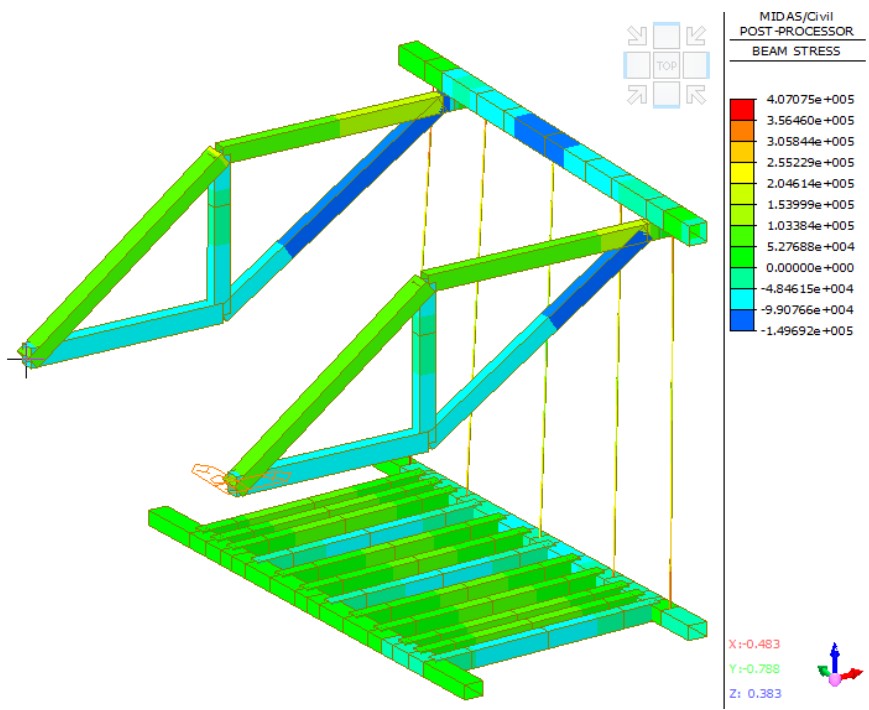

**Figure 19.** Stress nephogram of overall structure of hanging basket under condition 1.

1    Main Truss

The main truss is the main load-bearing member [23], which is composed of upper and lower chords (AB,CD), vertical web members (BD), tension diagonal web members (AD) and compression diagonal web members (BC), all of which are made of Q235. According to the finite element analysis results of Figures 20 and 21, the maximum distortion and maximum stress at the front end of the main truss are 12.53 mm and 149.69 MPa, respectively. The analysis summary of the maximum stress, axial force and distortion of each member of the main truss are shown in Table 5. Each member of the main truss met the design requirements of the project.

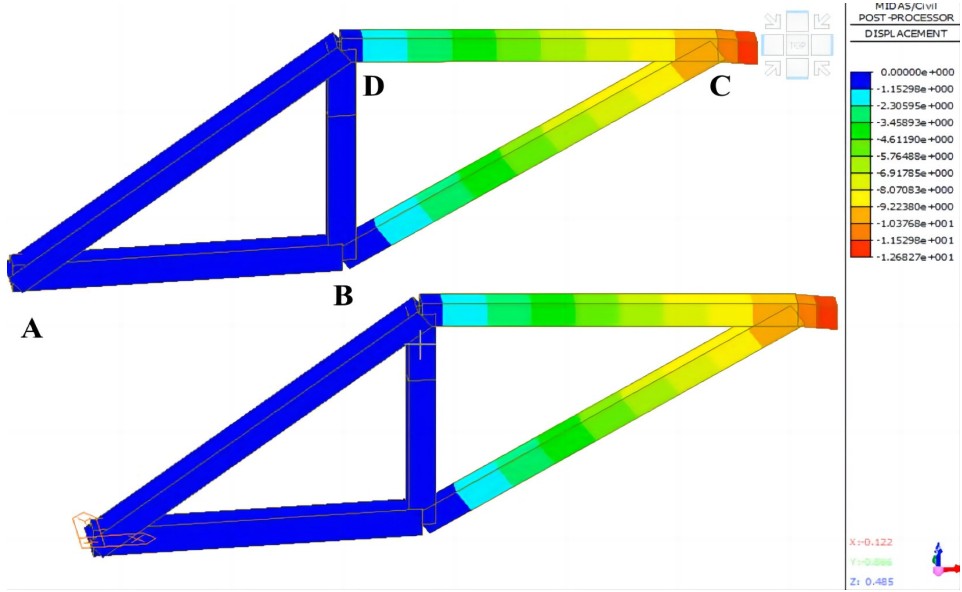

**Figure 20.** Distortion nephogram of main truss structure under condition 1.

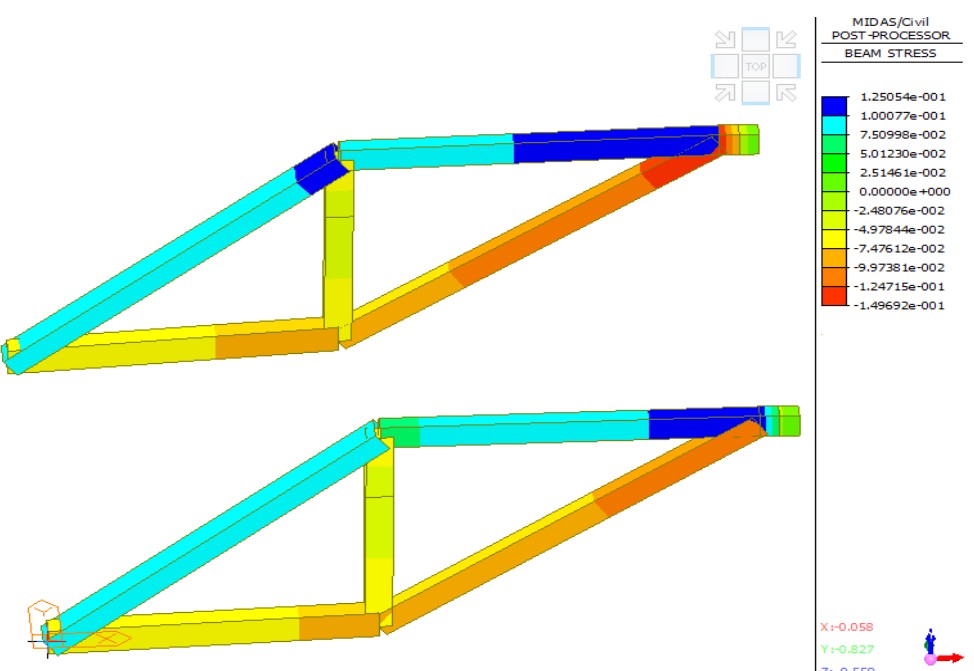

**Figure 21.** Stress nephogram of main truss structure under condition 1.

**Table 5.** Analysis of each member of main truss under condition 1.

| Member Name | Max Stress [MPa] | Max Vertical Distortion [mm] | Allowable Stress [MPa] | Allowable Distortion [mm] |
|---|---|---|---|---|
| AB | 92.6 | 0.25 | 175 | 11.75 |
| CD | 149.7 | 12.53 | 175 | 13.25 |
| BD | 55.0 | 0.41 | 175 | 9.00 |
| AD | 103.9 | 0.60 | 175 | 14.91 |
| BC | 134.2 | 10.33 | 175 | 15.25 |

2.  Front Upper Crossbeam

The front upper crossbeam is fixed at both ends of the upper chord of the main truss. According to the finite element analysis results from Figures 22 and 23, the maximum distortion in the middle of the front upper crossbeam is 25.00 mm. Due to the reference system, the actual maximum distortion and the maximum stress are 12.47 mm and 113.8 MPa, respectively.

3.  Bottom Basket

The bottom basket provided the bearing for the formwork and concrete pouring. According to the finite element analysis results of Figures 24 and 25, the maximum distortion of the bottom basket structure occurred in the suspender position in the middle of the front lower beam suspension, which is 28.65 mm. Due to the reference system, the actual maximum distortion of the front lower beam is 16.12 mm. In addition, the maximum stress occurred in the middle of the bottom longitudinal beam of the outermost web, which is 72.8 MPa. The maximum distortion and stress of the bottom basket structure met the engineering design requirements.

4.  Hanger Rod

The five hanger rods of the project are connected to the front upper beam and the front lower beam of the bottom basket. And the material is PSB785Φ32 finish rolled deformed steel bar. In order to ensure the engineering quality, each suspender can meet the design requirements. According to the finite element analysis results in Figures 26 and 27, the

maximum distortion of the No. 5 hanger rod at the front lower beam position is 16.11 mm, and the maximum stress at both ends of the No. 1 hanger rod is 407.07 MPa, which is less than the allowable stress of 650 MPa. The information about suspender distortion and stress is shown in Table 6.

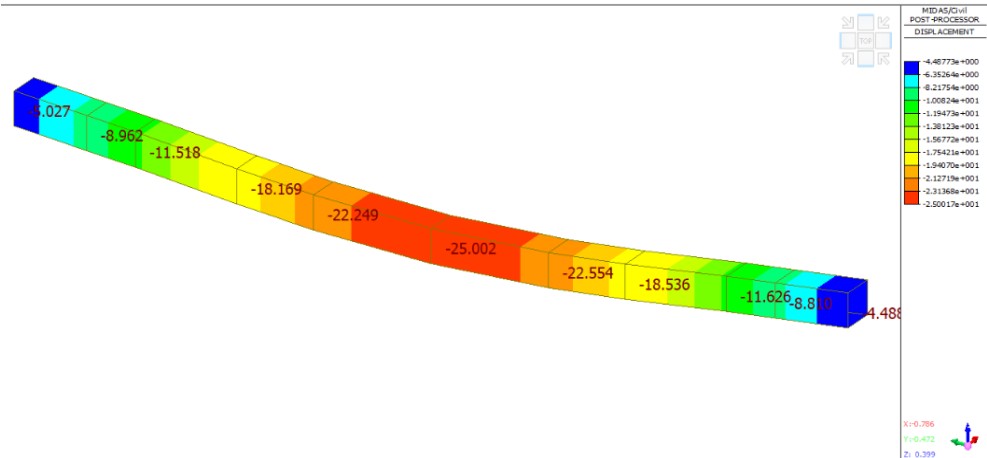

**Figure 22.** Structural distortion nephogram of front upper crossbeam under condition 1.

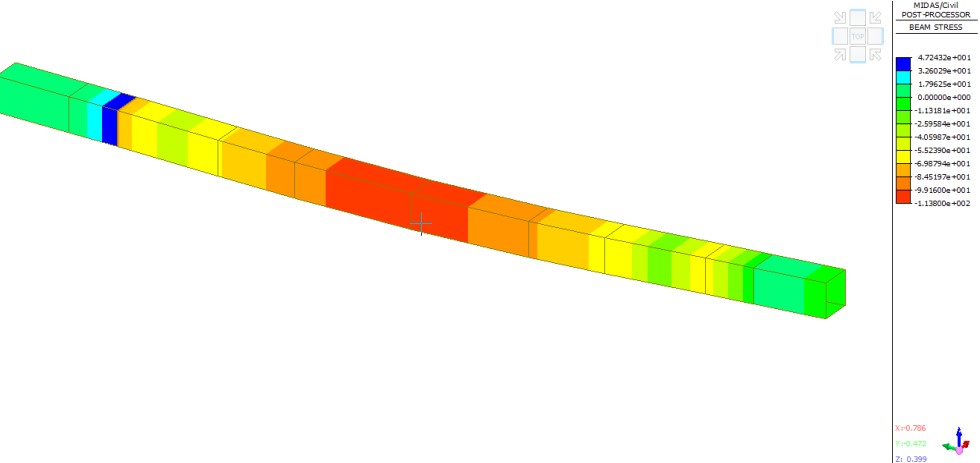

**Figure 23.** Stress nephogram of front upper crossbeam structure under condition 1.

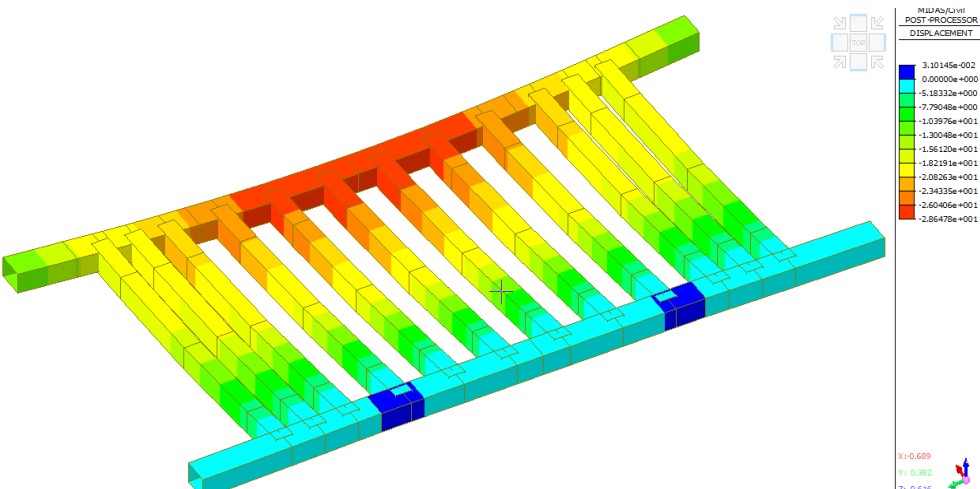

**Figure 24.** Cloud diagram of structural distortion of bottom basket under condition 1.

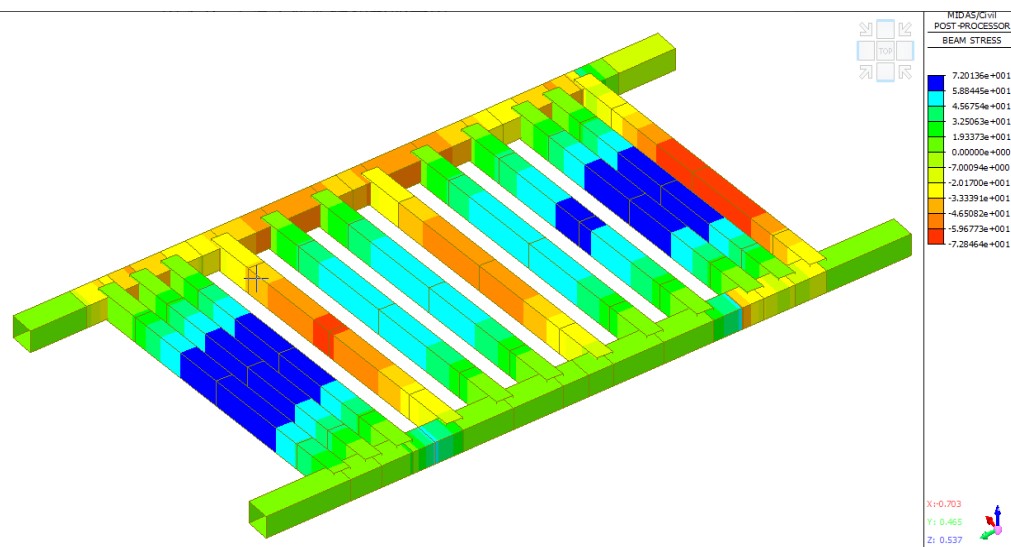

**Figure 25.** Stress nephogram of bottom basket structure under condition 1.

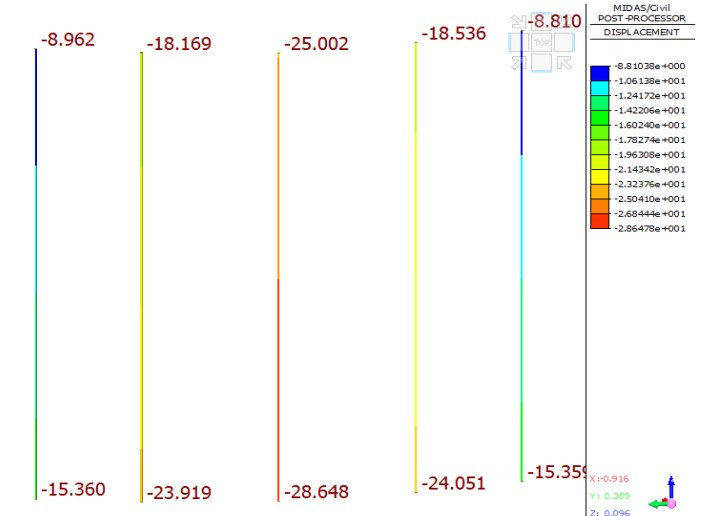

**Figure 26.** Nephogram of suspender distortion under condition 1.

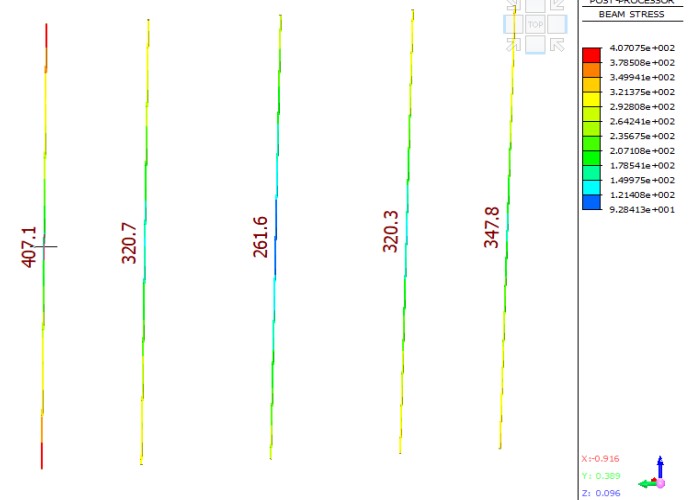

**Figure 27.** Nephogram of suspender stress under condition 1.

**Table 6.** Distortion and stress information of suspender in condition 1.

| Suspender Name | Max Distortion [mm] | Max Stress [MPa] | Allowable Stress [MPa] |
|:---:|:---:|:---:|:---:|
| 1 | 2.83 | 470.1 | 650 |
| 2 | 11.39 | 320.7 | 650 |
| 3 | 16.11 | 261.6 | 650 |
| 4 | 11.52 | 320.3 | 650 |
| 5 | 2.83 | 347.8 | 650 |

B    Analysis on Mechanical Performance of the Hanging Basket under Condition 2

In condition 2, the hanging basket walked to the end of the newly poured section, and it did not carry concrete, so the load it received was relatively small, but because it was in the last walking state, the telescopic sleeve of the lower chord of its main truss was fully extended. Therefore, the force of the structure still needed to be verified by simulation. After the hanging basket model of condition 2 was established in MIDAS/Civil software, load and boundary conditions were added. The overall distortion cloud map and stress cloud map of the structure of condition 2 are shown in Figures 28 and 29, respectively. It can be seen from Figure 28 that the maximum distortion of the structure of the hanging basket under working condition 2 was at the front end point of the right main truss, and its value is 10.70 mm, which is less than 13.25 mm of the length of the rod.

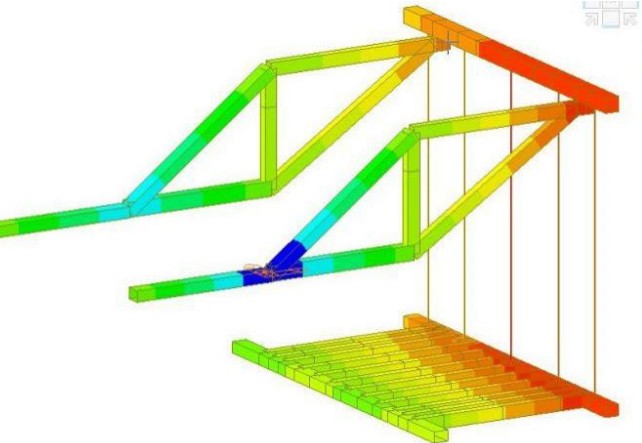

**Figure 28.** Cloud diagram of z-direction distortion of overall structure of hanging basket under condition 2.

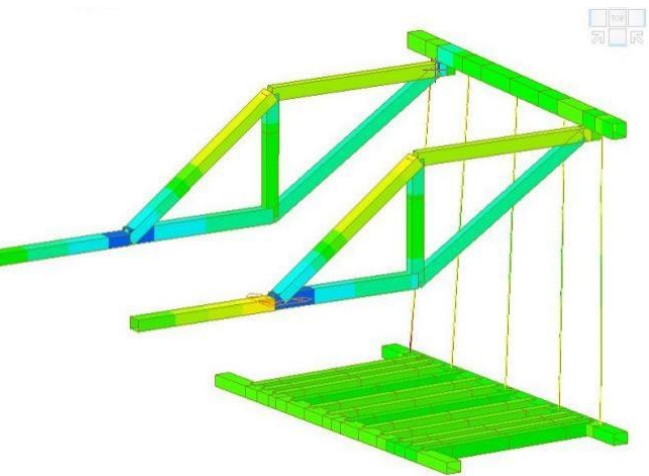

**Figure 29.** Cloud diagram of overall structural stress of hanging basket under working condition 2.

It can be seen from Figure 29 that the maximum stress of the structure of the hanging basket of working condition 2 was 137.1 MPa, which occurred at the position of the lower chord of the right main truss; a larger stress was generated, but the stress on the structure was less than the allowable stress of Q235 steel 175 MPa, which met the design requirements of the project. The main components of the hanging basket in working condition 2 are analyzed below.

1    Main truss

Different from condition 1, in addition to the original five members (upper and lower chords, vertical web members, tension sloping web members and compression sloping web members) of the main truss, the telescopic beam (AE) in the lower chord extended out. Due to the movement requirements, the rear anchor point untied the two flat load beams, leaving only one anchor for anchorage, resulting in a change in the load-bearing mode of the hanging basket. The material of each component is Q235. From the finite element analysis results of Figures 30 and 31, it can be known that the maximum distortion of the front end point of the main truss is 10.70 mm, and the maximum stress is 137.1 MPa.

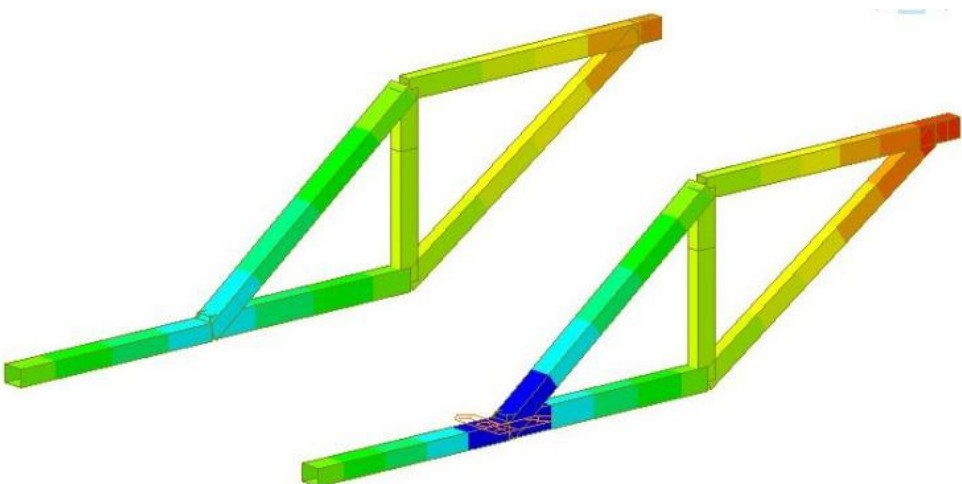

**Figure 30.** Cloud diagram of main truss structure distortion under condition 2.

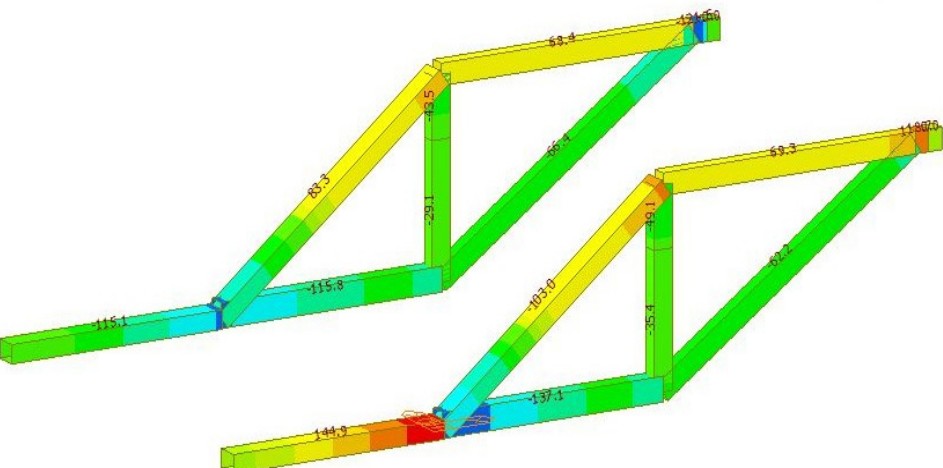

**Figure 31.** Cloud diagram of main truss structure distortion under working condition 2.

The maximum stress and distortion analysis of each member of the main truss is shown in Table 7. It can be seen from Table 7 that each member of the main truss meets the design requirements of the project.

**Table 7.** Analysis of the members of main truss under condition 2.

| Member Name | Maximum Stress | Maximum Vertical Distortion/mm | Allowable Stress/MPa | Allowable Distortion/mm |
| --- | --- | --- | --- | --- |
| AB | 137.1 | 16.9 | 175 | 11.75 |
| CD | 118.7 | 9.9 | 175 | 13.25 |
| BD | 49.1 | 0.12 | 175 | 9.00 |
| AD | 103.0 | 7.0 | 175 | 14.91 |
| BC | 62.2 | 9.7 | 175 | 15.25 |
| AE | 114.9 | 7.1 | 175 | 11.25 |

2. Front upper beam

In condition 2, the distortion and stress cloud diagrams of the front upper beam are shown in Figures 32 and 33. From the figures, the maximum distortion of the front upper beam occurred at the right end of the beam, and its real distortion is 12.96 mm. The maximum stress on the upper beam is 56.7 MPa.

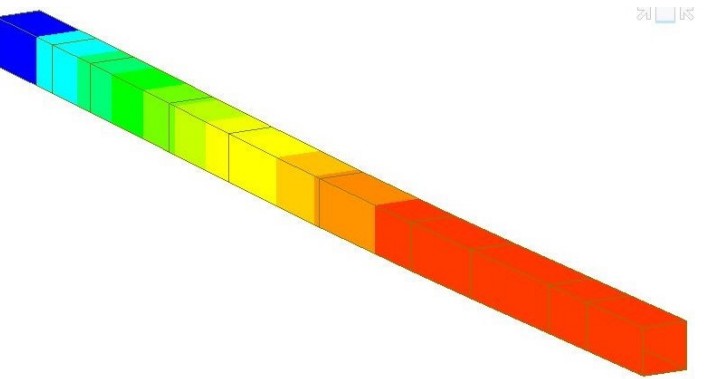

**Figure 32.** Cloud diagram of structural distortion of front upper beam under working condition 2.

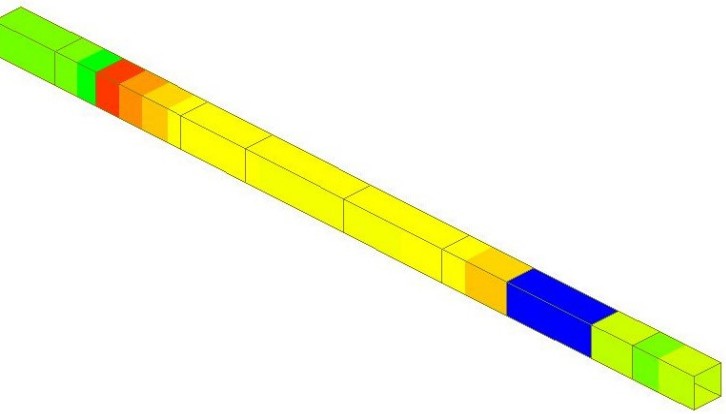

**Figure 33.** Nephogram of structural distortion of the front upper beam under condition 2.

3. Bottom basket

From the finite element analysis results of Figures 34 and 35, it can be seen that the maximum distortion generated in the bottom basket structure is 12.54 mm, which is located at the right end of the front lower beam, and the maximum stress is 93.36 MPa, which occurred in the rear lower beam. The structure of each part of the bottom basket, composed of the lower bottom longitudinal beam of the web plate, the lower bottom longitudinal beam of the bottom plate, and the front and rear lower transverse beams met the construction requirements.

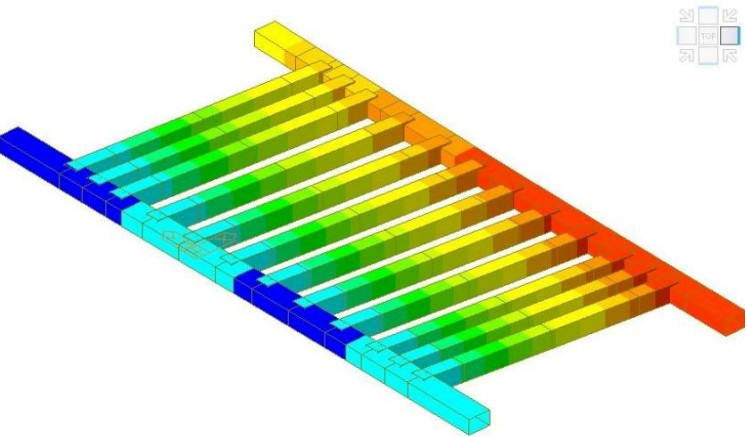

**Figure 34.** Cloud diagram of structural distortion of bottom basket under working condition 2.

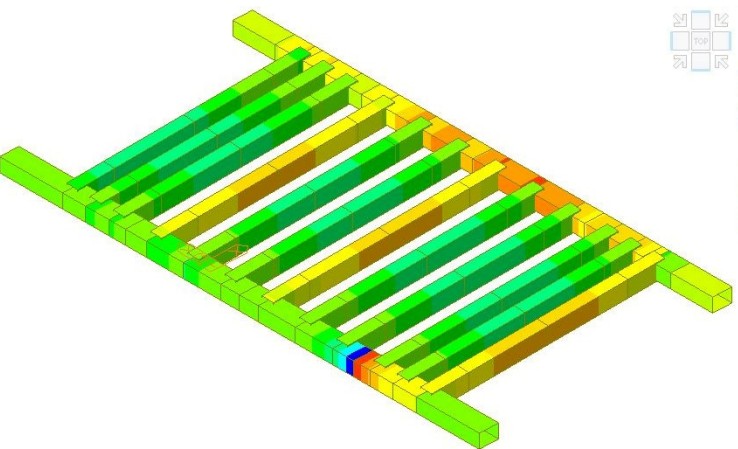

**Figure 35.** Cloud diagram of structural stress of bottom basket under working condition 2.

4.   Suspender

The finite element analysis results of the suspender in working condition 2 are shown in Figures 36 and 37, respectively. It can be seen from the figure that the maximum distortion of the structure occurred in the No. 5 suspender with a value of 23.18 mm, and the maximum stress occurred in the No. 1 suspender with a value of 295.4 MPa, which is less than the allowable stress of 650 MPa. The structural construction requirements, the distortion value and the stress of the suspender are shown in Table 8.

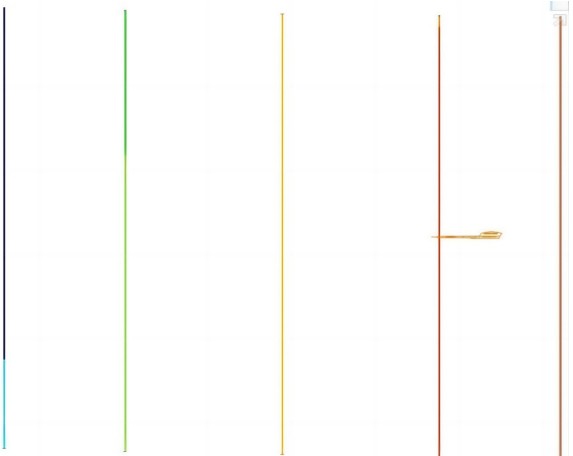

**Figure 36.** Cloud diagram of suspender structure distortion under working condition 2.

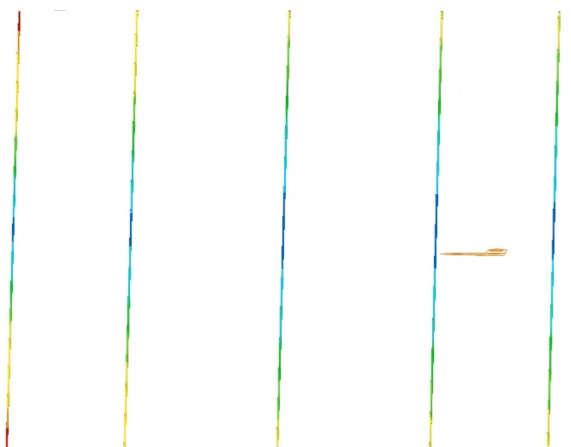

**Figure 37.** Cloud diagram of suspender structure stress under working condition 2.

**Table 8.** Suspender distortion and stress information under condition 2.

| Suspender Name | Maximum Distortion/mm | Maximum Stress/MPa | Allowable Stress/MPa |
|---|---|---|---|
| 1 | 17.13 | 295.4 | 650 |
| 2 | 19.37 | 218.8 | 650 |
| 3 | 21.76 | 260.1 | 650 |
| 4 | 22.94 | 162.3 | 650 |
| 5 | 23.18 | 167.9 | 650 |

C    Analysis of Mechanical Performance of the Hanging Basket under Condition 3

In order to analyze the mechanical performance of condition 3, it is indispensable to calculate its anti-overturning performance. In condition 3, the hanging basket is in the walking state, where the wind direction is vertically downward. According to Equation (2), $W_d$ represented wind load, $\gamma_a = 1.25$ kg/m$^3$ represented air density, $V_w = 13.6$ m/s represented wind speed. According to Equations (3) and (4), the dead weight load of the main truss is $G_1 = 63{,}000$ kg; dead weight load of the formwork is $G_2 = 63{,}000$ kg. In addition, the arms of force $X$, $X_1$ and $X_2$ are 5000 mm, 1000 mm and 2500 mm, respectively. In addition, the acceleration of gravity is $g$.

$$W_d = \frac{\gamma_a V_W}{2g} \tag{2}$$

$$M_{Q1} = G_1 g X_1 \tag{3}$$

$$M_{Q2} = G_2 g X_2 \tag{4}$$

Therefore, the total overturning moments of the hanging basket equipment are $M_{Q1}$, $M_{Q2}$ and $M_{Q3}$, which added up to $1.36 \times 106$ N·M; the anti-tipping moment of the equipment mainly came from the two rear walking trolleys. The ultimate bearing capacity of a single trolley is $3.5 \times 105$ N. And the stabilizing moment generated by the trolley is $q5 = 1$ kN/m, so the anti-overturning coefficient of the hanging basket is 2.57 (more than 2).

Consequently, the overall anti-overturning performance of the hanging basket in the walking condition met the operating requirements. Therefore, the overturning moment generated by the wind load is $M_{Q3} = W_d X$. The overturning moment generated by the dead weight of the main truss and the overturning moment generated by the dead weight of the formwork system are shown in Equations (3) and (4), respectively.

*2.4. Verification of Key Components and Analysis of the Weak Structures*

In this section, the distortion and stress of each part of the hanging basket structure are calculated, and compared with the results of the previous section. The applicability of

the finite element simulation method is verified. Ultimately, the paper put forward some suggestions for optimization of the distortion and stress of the main working conditions.

2.4.1. Member Calculation

The 9# block length is 4.5 m, and the segment volume is 79.6 m$^3$. According to the stress mode of the hanging basket mentioned above, the layout of the front lower beam and bottom longitudinal beam of the hanging basket is shown in Figure 38. In the calculation of this section, the concrete bulk density is $\gamma$ = 2.6 t/m$^3$, the overload coefficient of concrete the construction is $k_1$ = 1.05, the partial load coefficient is $k_2$ = 1.2, and L$'$ is the length of the current calculated beam section.

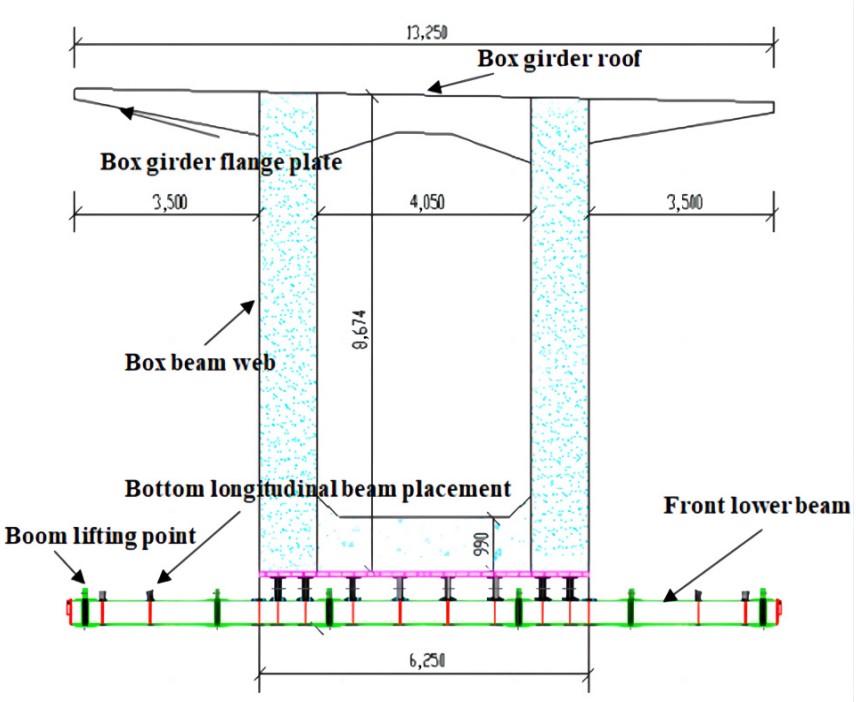

**Figure 38.** Front lower beam and bottom longitudinal beam.

A    Bottom Longitudinal Beam of Web

Take the bottom longitudinal beam as an example to calculate its deflection. The bottom longitudinal beam under the web and the bottom longitudinal beam under the bottom plate are calculated as simply supported beams. Because the bearing position and bearing capacity are different, they are treated separately for the calculation. If the section area of the box girder web is $s_1$ = 4.12 m$^2$, the load $q_1$ of the longitudinal beam under the 9# box girder web is shown in Equation (5).

$$q_1 = s_1\gamma k_1 k_2 = 135 \text{ kN/m} \tag{5}$$

The load of the external operation platform is $q_2$ = 2.5 kN/m$^2$, the dead weight of each longitudinal beam is $q_3$ = 0.33 kN/m, and the weight load of the upper bottom formwork of each longitudinal beam is $q_4$ =0.34 kN/m. The longitudinal beam under the web is calculated as a simply supported beam, and the calculation diagram is shown in Figure 39. There are two longitudinal beams under the 9# segment web, but only one is considered in the calculation, so the actual load qwb of the longitudinal beam under the web is shown in Equation (6).

$$q_{wb} = \frac{q_1}{2} + q_3 + q_4 = 68.52 \text{ N/mm} \tag{6}$$

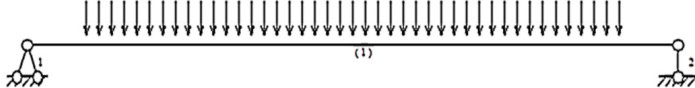

**Figure 39.** Calculation stress diagram of bottom longitudinal beam under web.

The reaction force of the two fulcrums of the bottom longitudinal beam under the web is 154.13 kN, the deflection of the bottom longitudinal beam under the web is shown in Figure 40, and the maximum deflection of the structure is 9.7 mm (less than 15.25 mm).

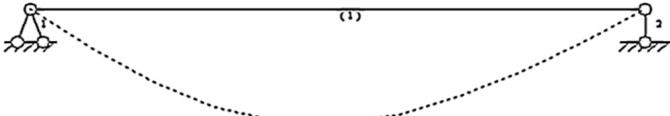

**Figure 40.** Deflection diagram of bottom longitudinal beam under web.

B    Bottom Longitudinal Beam under Bottom Plate

The longitudinal beam under the box girder bottom plate bore the load of the box girder bottom plate; the cross-sectional area of the box girder bottom plate is $s_2 = 3.12$ m². And the load of the longitudinal beam under the box girder bottom plate is shown in Equation (7). The weight of the upper bottom formwork of each longitudinal beam is $q_5 = 1$ kN/m, and the bottom longitudinal beam of the bottom plate is calculated as a simply supported beam. The calculation diagram is shown in Figure 41, and the actual load $q_{bb}$ of the bottom longitudinal beam of the bottom plate is shown in Equation (8). The reaction force of the two fulcrums of the bottom longitudinal beam under the bottom plate is 78.32 kN, the deflection of the bottom longitudinal beam under the bottom plate is shown in Figure 42, and the maximum deflection of the structure is 13.5 mm (less than 15.25 mm).

$$q_4 = s_2 \gamma k_1 k_2 = 102.2 \text{ kN/m} \tag{7}$$

$$q_{bb} = \frac{q_4}{2} + q_3 + q_5 = 34.81 \text{ N/mm} \tag{8}$$

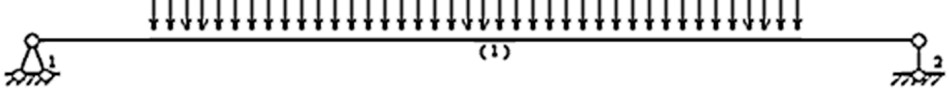

**Figure 41.** Calculation stress diagram of bottom longitudinal beam under bottom plate.

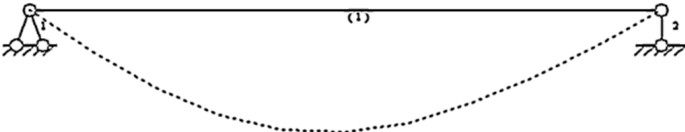

**Figure 42.** Deflection diagram of bottom longitudinal beam under bottom plate.

2.4.2. Analysis of the Weak Structure

In condition 1, the main components of the hanging basket structure are decomposed into a single element, the axial force, stress, and the strain of some main components are obtained. By comparing the calculated results with the calculated results of the previous section, it is found that the finite element analysis results are not much different from the theoretical calculation results. Therefore, it is reliable to calculate the stress of the hanging basket using MIDAS/Civil. In order to optimize the hanging basket more pertinently, the results obtained in the analysis and calculation are redundant with the allowable stress of the bar. The summary of the finite element calculation results and the allowable values of the members of the equipment are shown in Table 9. It can be seen from Table 9 that in the main structure of the hanging basket, the performance redundancy of the distortion of the

main truss CD bar, the bottom longitudinal beam under the floor, the front lower crossbeam and the stress of the main truss BC bar are less than 30%, while the performance redundancy of other bars is more than 30%. And all member materials are Q235. For this reason, the rigidity of the CD bar of the main truss, the lower bottom stringer of the bottom plate, the front lower beam, and the strength of the BC bar of the main truss should be focused on in subsequent research. Meanwhile, given that most bars have a high performance redundancy, the amount of carried concrete can be increased to extend the casting segment.

**Table 9.** Comparison of finite element calculation results and allowable values of members.

| Member | | Finite Element Analysis Results | | Allowable Value | | Performance Redundancy | |
|---|---|---|---|---|---|---|---|
| | | Stress [MPa] | Distortion [mm] | Stress [MPa] | Distortion [mm] | Stress [%] | Distortion [%] |
| Main trusses | AB | 92.6 | 0.21 | 175 | 11.75 | 47.09 | 98.21 |
| | AD | 103.9 | 0.60 | 175 | 14.91 | 40.63 | 95.98 |
| | BC | 134.2 | 10.33 | 175 | 15.25 | 23.31 | 32.26 |
| | BD | 55.0 | 0.41 | 175 | 9.00 | 68.57 | 95.44 |
| | CD | 119.7 | 12.53 | 175 | 13.25 | 31.60 | 5.43 |
| Front upper crossbeam | | 113.8 | 12.47 | 175 | 18.13 | 34.97 | 31.22 |
| Bottom longitudinal beam | Under the Web | 71.32 | 9.05 | 175 | 15.25 | 59.25 | 40.66 |
| | Under the floor | 57.99 | 13.67 | 175 | 15.25 | 66.86 | 10.36 |
| Lower cross beam | Front lower beam | 52.93 | 16.12 | 175 | 18.00 | 69.75 | 10.44 |
| | Lower rear beam | 48.59 | 9.80 | 175 | 18.00 | 72.23 | 45.56 |

## 3. Results

### 3.1. Main Truss Optimization

A    Front Elevation of Main Truss

The schematic diagram of the front elevation angle of the main truss of the hanging basket is shown in Figure 43, and the range of the front elevation angle can be taken as 0–35°. In this study, the initial front elevation was 35°, and the front elevation was gradually reduced to 5 degrees when the horizontal distance between the upper end of the main truss and the web member was fixed. According to the simulation results, the most suitable front elevation value was analyzed.

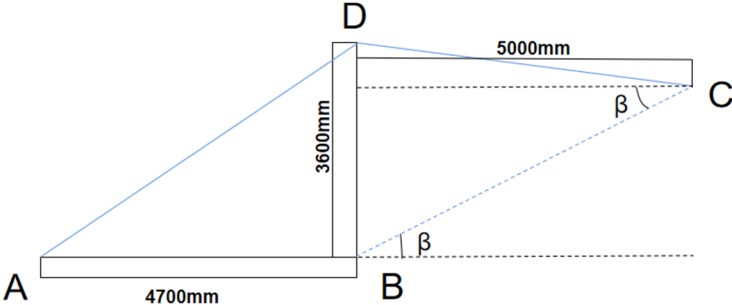

**Figure 43.** Front elevation of main truss.

The front angle of elevation of each simulation model was 5° lower than that of the previous model. Except for the model with a front angle of elevation of 35°, six additional groups of simulation models were established for the analysis of changes in bar length and steel consumption caused by the changes in the angle, as shown in Table 10 and

Figure 44. According to the analysis, relying on the previous finite element calculation, the load was applied. In the most unfavorable condition 1, at point C, the single main truss was subjected to the pressure of 399.32 t, which was respectively applied to the newly established six models. And the changes in stress in various bars are shown in Figure 45. Because the z-direction distortions of the BC and CD bars of the main truss are relatively large, the other three bars can be ignored.

**Table 10.** Model parameters for changing front elevation angle.

| Front Elevation Angle | 35° | 30° | 25° | 20° | 15° | 10° | 5° |
|---|---|---|---|---|---|---|---|
| BC + BD Pole length [m] | 11.161 | 10.824 | 10.675 | 10.628 | 10.664 | 10.768 | 10.935 |
| BC + BD Rod steel amount [kg] | 1421.5 | 1378.5 | 1359.6 | 1353.6 | 1358.1 | 1371.4 | 1392.7 |

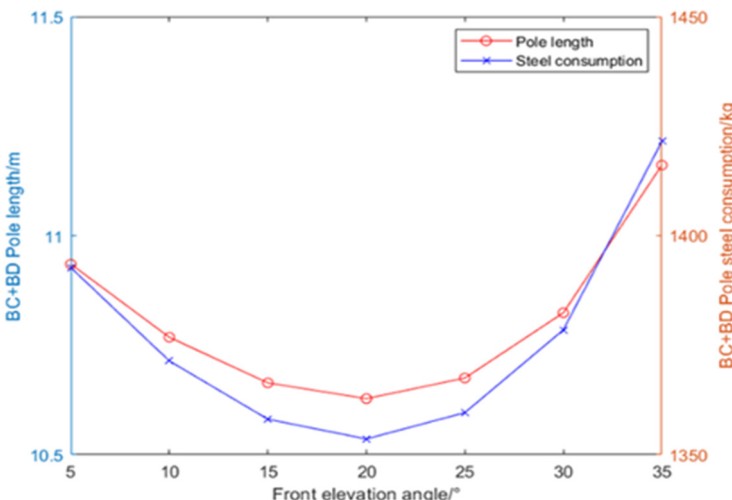

**Figure 44.** Schematic diagram of model parameters change caused by the change of front elevation angle.

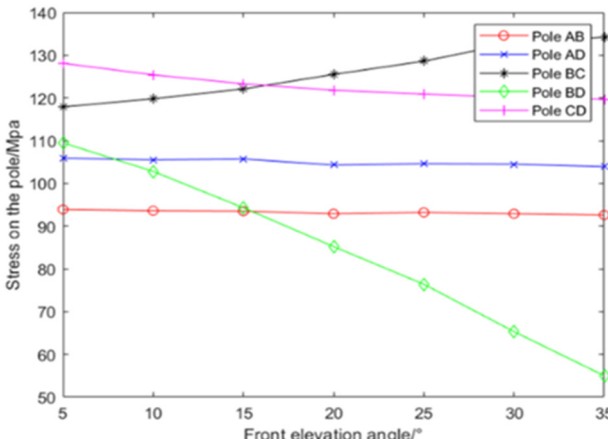

**Figure 45.** Stress variation of model member caused by change of front elevation angle.

The comparison of z-direction distortions in BC and CD bars is shown in Figure 46. It can be seen from the analysis of Figure 44 that when the horizontal distance between the upper end of the main truss and the web member is fixed, the front elevation angle of the main truss is reduced, and the steel consumption of the hanging basket is changed. In the range of a 5–35° front elevation, the sum of the BC and BD pole length is first decreased

and then increased. When the front elevation is 20°, the length of the BC + BD pole is the smallest, which is 10.628 m. And the steel consumption is the least, which is 1353.6 kg.

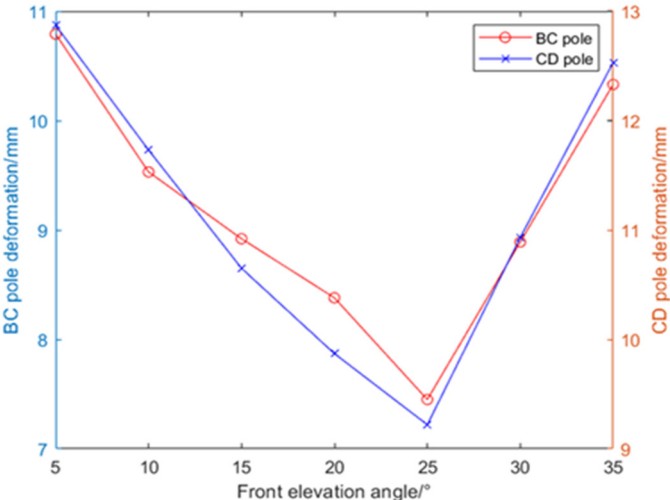

**Figure 46.** Distortion of important members of the model.

It can be seen from the analysis of Figure 45 that the stress changes of each member are different, and the stress changes of AB and AD poles are not obvious and can be ignored. In comparison, the stress of the BC, BD and CD bars witnessed changed significantly, in which the stress of BC bar was decreased with the reduction of the front angle of elevation, and the deceleration gradually eased off. As the BD bar belonged to the vertical belly pole, its stress was drastically increased with the reduction of the front angle of elevation. The stress of the CD bar was increased with the reduction of the front angle of elevation, and the deceleration gradually eased off. As can be seen from the analysis of Figure 46, the distortions of the BC and CD bars with a large vertical distortion of the main truss were first decreased and then increased to the reduction in the front angle of elevation. It can be observed that the changes were relatively small and the max distortion is still within the allowable distortion in the bar.

Based on the above simulation results and data, the optimal structure is selected when the front elevation angle is 20° and the steel consumption of BC + BD is 1353.6 kg. The stress of AB, AD, BC, BD and CD are 92.9, 104.3, 125.5, 85.2 and 121.8MPa, respectively. The vertical distortion of BC and CD poles are 8.38 mm and 9.87 mm, separately. The structure of the front elevation angle is shown in Figure 47, and the structure is substituted into the subsequent research for analysis.

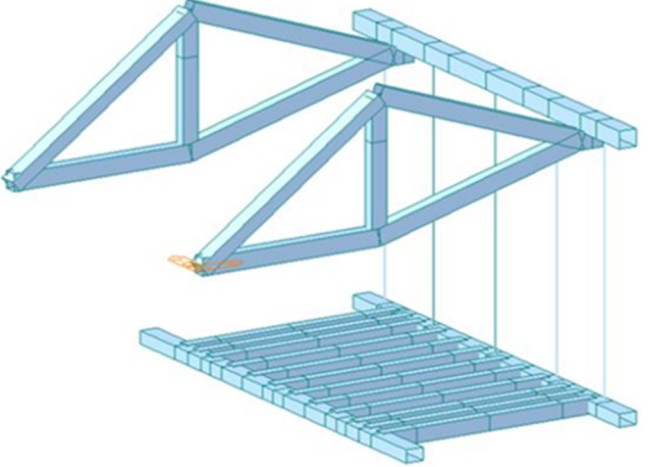

**Figure 47.** Hanging basket structure with a front elevation angle.

B    Main Truss Beam Structure

In the previous calculation, it is found that the mechanical performance of most members of the main truss have great redundancy. Therefore, in addition to optimizing the elevation angle and the mentioned extension of the pouring section, the beam structure can be changed; that is, the beam section can be changed to reduce the steel consumption of the equipment to improve the engineering efficiency. In the initial simulation calculation, each member of the main truss is made of steel, the vertical web member (BD) of the main truss is $350 \times 250 \times 12$ mm, the upper chord (CD), lower chord (AB) and compression diagonal web (BC) of the main truss are $400 \times 300 \times 12$ mm, the tension diagonal web member (AD) of the main truss is $420 \times 300 \times 12$ mm. Since it is found that the rigidity of the CD pole and the strength of the BC pole are close to the allowable stress in the calculation, only AB, AD and BD poles are optimized while maintaining the structure with the elevation angle of 20° before the upper section, keeping the length of BC and CD unchanged. The information on optimized poles is shown in Table 11. It can be seen from Table 11 that after optimization, the weight of AB, AD and BD poles is reduced by 14.976 kg, and the steel consumption is reduced by 29.952 kg because the hanging basket structure is symmetrical.

**Table 11.** Comparison table of optimized main truss member information.

|  | Original Dimension of Pole/mm | Optimized Pole Size/mm | Reduced Steel Consumption/kg |
|---|---|---|---|
| AB pole | $400 \times 300 \times 12$ mm | $380 \times 280 \times 12$ mm | 3.744 |
| AD pole | $420 \times 300 \times 12$ mm | $400 \times 280 \times 12$ mm | 7.488 |
| BD pole | $350 \times 250 \times 12$ mm | $330 \times 230 \times 12$ mm | 3.744 |

*3.2. Optimization of Pouring Section*

According to the previous calculation, it is found that the stiffness of the main truss except the CD member and the strength of the BC member have greater performance redundancy. In the actual construction, the single span is 100 m, the construction of the hanging basket is required to be divided into 24 segments for symmetrical casting, and the lengths of the segments are 3.5 m, 4 m and 4.5 m, respectively. In other words, the hanging basket needs to walk 12 times, of which there are two segments for every 3.5 m. In addition, four segments for 4 m and six segments for 4.5 m served as the heaviest segment, weighing at 206.96 t. In this section, the casting segment was extended by 0.5 m, while the heaviest segment was increased to 5 m in length and to 229.96 t in weight. The load value was applied to the optimal model established and affirmed in Section 3.1, so the distortion and stress nephogram of the hanging basket were obtained, as shown in Figures 48 and 49.

It can be seen from Figure 48 that the vertical distortion of 13.14 mm occurred at the upper end of the main truss, which is less than 1/400 of the length of the pole (13.27 mm); the vertical displacement of 29.55 mm occurred in the middle of the front upper beam, which is less than 1/400 of the length of the pole (30 mm) due to the reference system. In the bottom basket structure, the vertical displacement of 33.09 mm occurred in the middle of the front lower beam, which is due to the reference system. According to different reference systems, the actual distortion of the pole is 19.95 mm; the distortion of the lower part of the No. 3 suspender is 19.95 mm. The distortion of the structure met the design requirements. As can be seen from Figure 49, except the suspender, the maximum beam unit internal force generated by the whole structure is 161.19 MPa, which occurred at the junction of the front upper beam and the main truss. The structural stress is less than the allowable stress of 175MPa of Q235 steel. $\left[\sigma_{Q235}^a\right] = 175$ MPa. The maximum stress at both ends of the No. 1 suspender is 406.17 MPa, which is less than the allowable stress of 650 MPa of the finish rolled rebar and met the design requirements of the project.

In this section, first of all, the simulation results showed that when the front elevation angle is 20°, the steel consumption of the main truss is the least, and under the condition of improving the front elevation angle, the optimization of the beam structure can reduce

the steel consumption of the two main trusses by 29.952 kg. Secondly, when the optimized main truss structure is determined, the pouring section is extended by 0.5 m. After the finite element simulation of the new construction scheme, it was found that the stress of the hanging basket structure after the extension of the pouring section still met the engineering requirements. The maximum deformation of the main truss structure is 13.14 mm and the maximum stress is 161.19 MPa. The optimization of the structure and the construction saved engineering resources and costs.

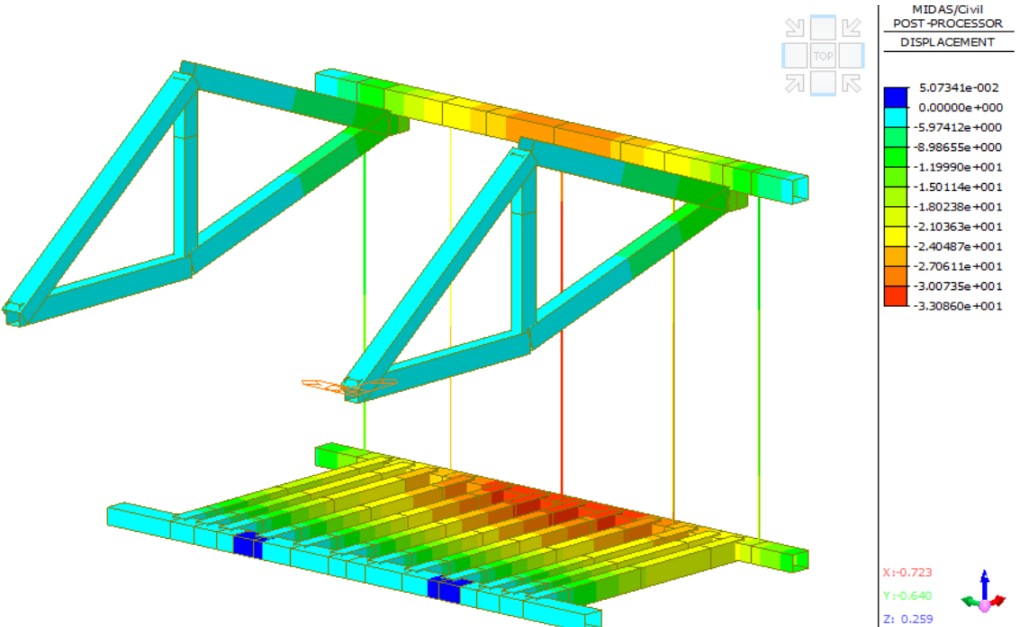

**Figure 48.** Z-direction distortion nephogram of overall optimized structure of hanging basket for extended casting segment.

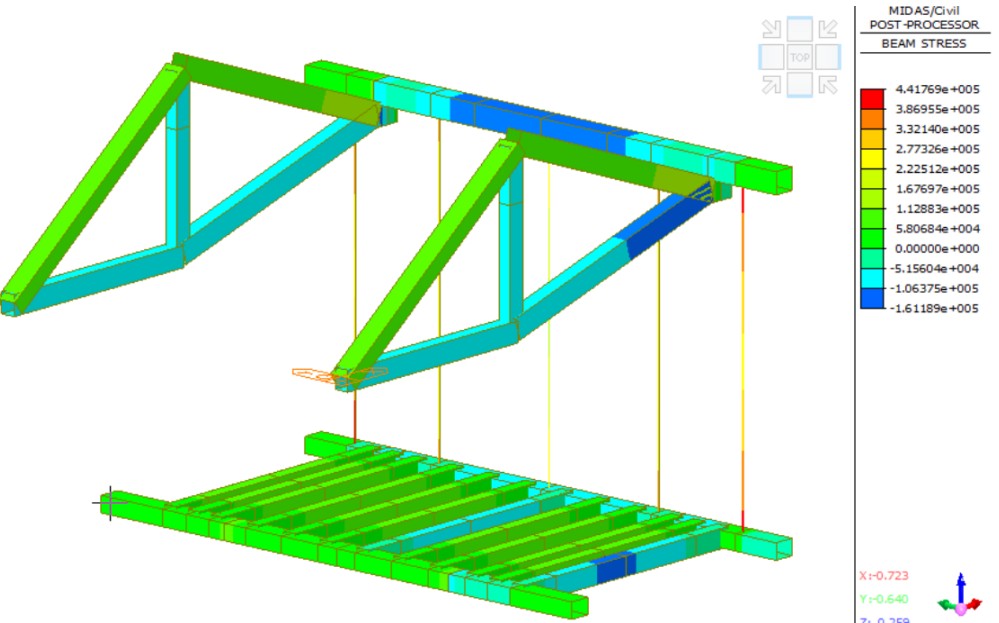

**Figure 49.** Stress nephogram of overall optimization structure of hanging basket for extended casting segment.

## 4. Conclusions

The traditional hanging basket walking mechanism is to lay steel rails on the box girder working panels, which are fixed by front and rear supports and prestressed steel bars. The height of the steel rails is higher, which increased the overall center of gravity of the equipment and raised the overturning of the hanging basket during the construction. In addition, the walking track is long and heavy, and the process of paving and dismantling is complicated, which limited the improvement of the efficiency of the bridge construction. Relying on the Tianjin Hai-he Bridge, this paper designed a novel rhombus traveling track of a hanging basket, which has higher tensile strength, higher compressive strength and better impact resistance than traditional rails. The design and analysis of the force state of the rhombus hanging basket running under different working conditions, and the finite element simulation and calculation were carried out; based on the simulation and calculation results, the suitable hanging basket transformation for the project is proposed and verified. The scheme optimized the structure of the hanging basket components, reduced weight and saved costs, shortened the construction period, and improved the construction efficiency.

Using business-oriented software MIDAS/Civil, the mechanical properties of the rhombus hanging basket under three working conditions, fully loaded and unloaded, are analyzed. In working condition 1, the maximum beam unit internal force of 149.69 MPa occurred at the joint between the front upper beam and the main truss, which is less than the allowable stress of Q235 steel of 175 MPa. In working condition 2, the maximum stress occurred in the No. 1 suspender with a value of 295.4 MPa, which is less than the allowable stress of 650 MPa. In working condition 3, the anti-overturning safety factor of the hanging basket is 2.5 to meet the requirements, and the strength and distortion of the walking mechanism are within the specified range; the key components are calculated and compared with the finite element calculation, and it is found that the finite element calculation is reliable and the structure performance is 30% with left and right redundancy. In order to further improve the utilization efficiency of the suspended pouring basket, the main truss and the pouring stage are optimized. It is found that when the front elevation angle of the main truss is 20°, the steel consumption is the least, the member stress is small, and the stress of each member is relatively uniform and there is small vertical distortion; when the pouring section is extended by 0.5 m, the heaviest section is extended to 5 m, and when the weight is increased to 229.96 t, the working condition still meets the requirements. The research results of this paper can provide references for the structural optimization and improvement of the rhombus hanging basket and similar cantilever construction projects.

**Author Contributions:** Y.O. had management and coordination responsibility for the research activity planning and execution. J.H. and K.S. designed the experiments and J.H. carried them out. K.S. used the civil software and performed the simulations. J.H. prepared the manuscript with contributions from all co-authors. All authors have read and agreed to the published version of the manuscript.

**Funding:** This paper is supported by the Key research and development project of Jiangxi Province: Research and application of Key Technologies of intelligent Management System of hilly and mountainous orchard Transport Equipment based on "Internet +" (20202BBF63016); Key research and application of Key Technologies of efficient intelligent Freight System of mountainous orchard (20212BBF63041). The authors are grateful for the support.

**Institutional Review Board Statement:** Not applicable.

**Informed Consent Statement:** Not applicable.

**Data Availability Statement:** The data used to support the findings of this study are included within the article.

**Conflicts of Interest:** The authors declare that they have no conflict of interest.

## Nomenclature

| | |
|---|---|
| $a_{cU}$ | The impact strength of the specimen $(kJ/m^{2)}$ |
| $h_{cU}$ | The thickness of the specimen (mm) |
| $E_{cU}$ | The energy absorbed by the measured pattern (J) |
| $b_{cU}$ | The width of the specimen (mm) |
| $W_d$ | The wind load |
| $\gamma_a$ | The air density $(kg/m^{3)}$ |
| $V_w$ | The wind speed (m/s) |
| $g$ | The acceleration of gravity |
| $M_Q$ | The overturning moment of the hanging basket equipment (N·m) |
| $G$ | The dead weight load (kg) |
| $X$ | The arm of force (mm) |
| $q$ | The load of structure |
| $s$ | The cross-sectional area of structure $(m^2)$ |
| $\gamma$ | The concrete bulk density $(t/m^{3)}$ |
| $k$ | The load coefficient |

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
