# Peer review of "Development of Rhombus Hanging Basket Walking Track Robot for Cantilever Casting Construction in Bridges"

_applsci, doi:10.3390/app131910635_

Round 1

Reviewer 1 Report

This paper presents the development of a Hanging Basket walking track for cantilever casting to construct bridges. The title of the manuscript refers to the development of the robot, however, the robotic automation part was not addressed in the manuscript. However, the authors proposed a new type of walking track that addresses some of the issues previously had by similar walking tracks. The authors also discuss the materials used to develop the walking tracks and show the results of material testing. The reviewer has the following comments:

  • Considerable work is needed to improve the written quality of the paper in terms of grammar and flow of sentences in particular.
  • Length of the manuscript is too long. The authors are required to considerably reduce the paper length by deleting unnecessary text.
  • Novelty and significance of the work is not clear.
  • Figure 3 and other figures' text should be increased in font size, to make it readable.
  • No need to put “ï‚£” symbol in the text. With the Figure it is OK
  • MIDAS is a commercially available software, which must be mentioned in the text
  • Why ABAQUS is used when ANASYS is a more powerful tool for material simulation.
  • “Mpa” to be written as “MPa”.
  • Figure 36 & 37 needs to be labeled properly.
  • Legends of the figures are not clear.
  • Similarly, the stress contours shown in all of the Figures are not readable. The reviewer recommends regenerating the labels on the contours so that it becomes readable.
  • The conclusion part of the manuscript is missing.
  • The references are not as per the journal’s format.

Considerable work is needed to improve the written quality of the paper in terms of grammar and flow of sentences in particular.

Author Response

Dear reviewer, thank you for criticizing and correcting our manuscript in your busy schedule, and we have made the following changes in response to the issues you raised:

Considerable work is needed to improve the written quality of the paper in terms of grammar and flow of sentences in particular.

         We carefully proofread the article, making changes at some confusing statements and focusing on fixing tenses

Length of the manuscript is too long. The authors are required to considerably reduce the paper length by deleting unnecessary text.

          The content of the manuscript has been reduced as much as possible, but it is not possible to make large deletions without making logical sense of it.

Novelty and significance of the work is not clear.

           Relying on the Tianjin Haihe Bridge project, this paper proposes and designs a travel track for optimizing the movement of hanging baskets, and introduces its installation and operation mode, which can provide a reference for the optimization and improvement of the structure of diamond-shaped hanging baskets as well as similar cantilever construction projects.

Figure 3 and other figures' text should be increased in font size, to make it readable.

          The text in the image has been labeled as clearly as possible, and due to the size of the image, it is not possible to set its scale to the maximum.

No need to put “£” symbol in the text. With the Figure it is OK

        We  have deleted all “£” in the text

MIDAS is a commercially available software, which must be mentioned in the text

          It has already been done in the article

Why ABAQUS is used when ANASYS is a more powerful tool for material simulation.

          ABAQUS has some features that ANASYS does not have, and we have been using ABAQUS to follow up on this project

“Mpa” to be written as “MPa”.

All “Mpa” have been written as “MPa”.

Figure 36 & 37 needs to be labeled properly.

Figure 36 & 37 needs has been labeled properly.

Legends of the figures are not clear

          We're sorry we're not sure which numbers you're pointing out, but have changed some of them

Similarly, the stress contours shown in all of the Figures are not readable. The reviewer recommends regenerating the labels on the contours so that it becomes readable

          Pretty sorry, we don't have the conditions to re-run the experiment at the moment, we can't modify the problem you mentioned for now.

The conclusion part of the manuscript is missing

Conclusions are included in the discussion

The references are not as per the journal’s format。

References have been revised as required

Thanks again for all you do

Reviewer 2 Report

Dear Authors,

I have carefully examined your manuscript titled “Development of Rhombus Hanging Basket Walking Track Robot for Cantilever Casting The construction in Bridges " submitted to Applied Sciences. The work presented in the paper explores a novel approach to enhance the efficiency of cantilever casting in bridge construction by developing and optimizing a rhombus traveling track for a hanging basket. Overall, the research is engaging and relevant to the field of bridge engineering.

The research presented in the paper appears to be both relevant and interesting for the field of bridge construction and engineering. The development of a novel rhombus traveling track for hanging baskets to enhance the efficiency of cantilever casting is a promising concept that may have practical applications in real-world construction projects.

The use of the civil software Midas for analyzing the mechanical properties of the hanging basket demonstrates a comprehensive approach to evaluating the performance of the structure under various loads. This software is widely recognized and accepted in the civil engineering community, further adding credibility to the study.

Based on the thorough evaluation of the paper, it is noted that Figures 1,2 and 3 appear to have limitations in terms of export quality. Especially in the case of Figure 1, the visualization is compromised, and the details are not adequately conveyed.

Furthermore, in Figure 2, the presence of units directly alongside the dimensions may introduce confusion for readers. To ensure clarity and adherence to best practices in technical drawing, I highly recommend following the guidelines set forth in the ISO 129-1:2021 standard for dimensioning and tolerancing in technical drawings. Separating the units from the dimensions as per the ISO norms will greatly enhance the comprehensibility of the drawings and improve the overall quality of the paper.

About the tensile property and compressive performance tests: To provide reliable and meaningful results in tests, it is not sufficient to present a single value of tension. It is essential to include both the mean and the standard deviation of the data obtained during the experiments. The main reason behind this is to ensure accuracy and representativeness of the results. Presenting only a single value of tension in test results can be misleading and may not adequately reflect the variability or precision of the data. Therefore, it is crucial to include both the mean and the standard deviation to provide robust, reliable, and meaningful results that properly support the research and allow for a more comprehensive understanding of the phenomena studied.

In figures 7 and 10, the units are enclosed within parentheses in the axis labels. This formatting choice is made to avoid confusion with a division symbol represented by a forward slash (/). By using parentheses, it clarifies that the units are separate from the numerical values and are not intended to be part of a division operation. This practice ensures that the axis labels are presented in a clear and unambiguous manner, allowing readers to interpret the data accurately and without any misconceptions related to unit representation.

I would like to express that Figure 13 appears to be unclear and challenging to comprehend due to the presence of various colors and different styles of dimensioning. The combination of these elements makes the figure illegible, hindering the reader's ability to interpret the information effectively. To ensure the proper understanding of the data presented in Figure 13, I would kindly suggest reconsidering the visualization approach, simplifying the color scheme, and adhering to consistent dimensioning styles to enhance clarity and readability. This adjustment will significantly improve the overall quality of the figure and contribute to a better understanding of the findings.

In the case of Figures 17 to 27 and 48-49, it is challenging to read the color code due to its excessively small size. The color key or legend provided in the figure is not easily discernible, which can lead to confusion and difficulty in interpreting the data. To ensure the figure's clarity and to facilitate a better understanding of the color-coded information, I would recommend enlarging the color code or legend to a more readable size. This adjustment will greatly improve the figure's legibility and enable readers to identify and comprehend the represented data accurately.

In the context of Figures 36 and 37, it is evident that the lack of explanation has resulted in an unclear representation of the data. The figures appear to contain blurry lines and indistinct colors, making it challenging for readers to discern the information being conveyed. To address this issue, I highly recommend providing detailed explanations or captions that describe the content and purpose of each figure. Additionally, consider enhancing the visual quality of the figures by improving line clarity and choosing color schemes that offer better contrast and clarity. By incorporating these improvements, the figures will become more comprehensible and effectively convey the relevant data to the readers.

Regarding Figure 44, there appears to be a significant confusion with the color code, particularly distinguishing between blue and orange. It is not clear which data points or elements in the figure correspond to each color. To enhance the clarity and understanding of the figure, I recommend providing a clear and explicit color key or legend that clearly indicates the meaning of each color. Additionally, consider using colors that have a distinct contrast to avoid any ambiguity in interpreting the data. By addressing this issue, readers will be able to correctly identify and comprehend the data presented in the figure, leading to a more meaningful representation of the research findings.

The bibliographic references indeed appear to have significant spacing issues, and it seems that they are not properly formatted according to the journal's guidelines. The spacing between references appears excessive, which may not comply with the specific formatting requirements set by the journal.

I must express my concern regarding two main aspects that need improvement before considering publication.

Firstly, the quality of the graphics requires attention, as pointed out in several comments. The current graphical representation is subpar, with some figures being confusing and illegible. Clear and well-presented graphics are crucial for conveying the research findings effectively and enhancing the overall quality of the article.

Secondly, the presentation of experimental results needs to be revisited. The current approach lacks sufficient detail and clarity, making it challenging for readers to comprehend the outcomes fully. Providing additional data and refining the representation of results will enhance the impact and value of the research.

Given these important points, I recommend a thorough revision of the manuscript prior to its publication. Addressing these issues will elevate the overall readability and impact of the paper, ultimately contributing to its significance in the field.

I appreciate the effort and dedication invested in this research and believe that with the necessary improvements, the article will make a valuable addition to the scientific literature.

I trust that the provided feedback will be valuable for improving the manuscript. Please consider these suggestions carefully when revising the paper. Thank you for contributing your research, and I look forward to seeing the revised version in due course.

Sincerely,

PhD Reviewer

Author Response

Dear reviewer, thank you for criticizing and correcting our manuscript in your busy schedule, and we have made the following changes in response to the issues you raised:

Based on the thorough evaluation of the paper, it is noted that Figures 1,2 and 3 appear to have limitations in terms of export quality. Especially in the case of Figure 1, the visualization is compromised, and the details are not adequately conveyed.

We tried to scale the size of Figure 1, but its internal structure cannot be changed without affecting the data distribution

Furthermore, in Figure 2, the presence of units directly alongside the dimensions may introduce confusion for readers. To ensure clarity and adherence to best practices in technical drawing, I highly recommend following the guidelines set forth in the ISO 129-1:2021 standard for dimensioning and tolerancing in technical drawings. Separating the units from the dimensions as per the ISO norms will greatly enhance the comprehensibility of the drawings and improve the overall quality of the paper.

Sorry, this image is from an in-house design drawing and we can't alter it too much

About the tensile property and compressive performance tests: To provide reliable and meaningful results in tests, it is not sufficient to present a single value of tension. It is essential to include both the mean and the standard deviation of the data obtained during the experiments. The main reason behind this is to ensure accuracy and representativeness of the results. Presenting only a single value of tension in test results can be misleading and may not adequately reflect the variability or precision of the data. Therefore, it is crucial to include both the mean and the standard deviation to provide robust, reliable, and meaningful results that properly support the research and allow for a more comprehensive understanding of the phenomena studied.

The data we recorded during the experiment was provided by the relevant testing organization, and we are sorry that we could not ask for more and did not have the first-hand data.

I would like to express that Figure 13 appears to be unclear and challenging to comprehend due to the presence of various colors and different styles of dimensioning. The combination of these elements makes the figure illegible, hindering the reader's ability to interpret the information effectively. To ensure the proper understanding of the data presented in Figure 13, I would kindly suggest reconsidering the visualization approach, simplifying the color scheme, and adhering to consistent dimensioning styles to enhance clarity and readability. This adjustment will significantly improve the overall quality of the figure and contribute to a better understanding of the findings.

The purpose of our use of two colors was originally to more visually reflect the force on the bracket, Figure 13 has been changed to the original style

In the case of Figures 17 to 27 and 48-49, it is challenging to read the color code due to its excessively small size. The color key or legend provided in the figure is not easily discernible, which can lead to confusion and difficulty in interpreting the data. To ensure the figure's clarity and to facilitate a better understanding of the color-coded information, I would recommend enlarging the color code or legend to a more readable size. This adjustment will greatly improve the figure's legibility and enable readers to identify and comprehend the represented data accurately.

I apologize for the legend problem, we tried to change the angle of the graphs but that resulted in some obscure deformations not being shown, we tried to arrange the graphs as closely as possible to show the full amount of deformations.In order to visualize the model as a whole we had to compress the size of some of the legends, please understand the inconvenience caused.

In the context of Figures 36 and 37, it is evident that the lack of explanation has resulted in an unclear representation of the data. The figures appear to contain blurry lines and indistinct colors, making it challenging for readers to discern the information being conveyed. To address this issue, I highly recommend providing detailed explanations or captions that describe the content and purpose of each figure. Additionally, consider enhancing the visual quality of the figures by improving line clarity and choosing color schemes that offer better contrast and clarity. By incorporating these improvements, the figures will become more comprehensible and effectively convey the relevant data to the readers.

Figures 36, 37 are incorrectly labeled and has been labeled properly.

Regarding Figure 44, there appears to be a significant confusion with the color code, particularly distinguishing between blue and orange. It is not clear which data points or elements in the figure correspond to each color. To enhance the clarity and understanding of the figure, I recommend providing a clear and explicit color key or legend that clearly indicates the meaning of each color. Additionally, consider using colors that have a distinct contrast to avoid any ambiguity in interpreting the data. By addressing this issue, readers will be able to correctly identify and comprehend the data presented in the figure, leading to a more meaningful representation of the research findings.

Fig. 44 Using dual axes, the left side shows blue data and the right side shows orange data

The bibliographic references indeed appear to have significant spacing issues, and it seems that they are not properly formatted according to the journal's guidelines. The spacing between references appears excessive, which may not comply with the specific formatting requirements set by the journal.

References have been revised as necessary

Thanks again for all you do

Round 2

Reviewer 1 Report

I've had the opportunity to review the revised manuscript, and I appreciate the effort that has gone into addressing the previous comments and suggestions. However, I'd like to request that all revisions and corrections made in the manuscript be clearly marked or highlighted.

As per the instructions provided by the editorial office, it is crucial for reviewers to be able to identify and assess the changes made in the revised version easily. Unfortunately, without such markings, it is challenging for me to verify whether the requested revisions have been appropriately incorporated.

Therefore, I kindly ask that the authors ensure all modifications are clearly indicated, which will enable me to provide a more accurate and comprehensive review. If the necessary adjustments are made to the manuscript in this manner, I will be able to provide more meaningful feedback.

Thank you for your attention to this matter, and I look forward to reviewing the revised manuscript once the requested changes have been implemented.

I've had the opportunity to review the revised manuscript, and I appreciate the effort that has gone into addressing the previous comments and suggestions. However, I'd like to request that all revisions and corrections made in the manuscript be clearly marked or highlighted.

As per the instructions provided by the editorial office, it is crucial for reviewers to be able to identify and assess the changes made in the revised version easily. Unfortunately, without such markings, it is challenging for me to verify whether the requested revisions have been appropriately incorporated.

Therefore, I kindly ask that the authors ensure all modifications are clearly indicated, which will enable me to provide a more accurate and comprehensive review. If the necessary adjustments are made to the manuscript in this manner, I will be able to provide more meaningful feedback.

Thank you for your attention to this matter, and I look forward to reviewing the revised manuscript once the requested changes have been implemented.

Author Response

Dear reviewer, thank you again for your review, and in response to the improvements in the manuscript, we have red-flagged most of the changes, which are listed below:

Considerable work is needed to improve the written quality of the paper in terms of grammar and flow of sentences in particular.

         We carefully proofread the article, making changes at some confusing statements and focusing on fixing tenses.

Length of the manuscript is too long. The authors are required to considerably reduce the paper length by deleting unnecessary text.

          The content of the manuscript has been reduced as much as possible, but it is not possible to make large deletions without making logical sense of it.

Novelty and significance of the work is not clear.

           Relying on the Tianjin Haihe Bridge project, this paper proposes and designs a travel track for optimizing the movement of hanging baskets, and introduces its installation and operation mode, which can provide a reference for the optimization and improvement of the structure of diamond-shaped hanging baskets as well as similar cantilever construction projects.

Figure 3 and other figures' text should be increased in font size, to make it readable.

          The text in the image has been labeled as clearly as possible, and due to the size of the image, it is not possible to set its scale to the maximum.

No need to put “£” symbol in the text. With the Figure it is OK

        We have deleted all “£” in the text.

MIDAS is a commercially available software, which must be mentioned in the text

          It has already been done in the article.

Why ABAQUS is used when ANASYS is a more powerful tool for material simulation.

          ABAQUS has some features that ANASYS does not have, and we have been using ABAQUS to follow up on this project. The exact reason has been labeled in the article.

“Mpa” to be written as “MPa”.

           All “Mpa” have been written as “MPa”.

Figure 36 & 37 needs to be labeled properly.

           Figure 36 & 37 needs has been labeled properly.

Legends of the figures are not clear.

          We're sorry we're not sure which numbers you're pointing out, but have changed some of them.

Similarly, the stress contours shown in all of the Figures are not readable. The reviewer recommends regenerating the labels on the contours so that it becomes readable.

         We have re-generated all the images that contain stress contours to make the numbers on the images look clearer.

The conclusion part of the manuscript is missing.

        Conclusion has been added at the end of the article and labeled with the significance of the relevant research elements done in this paper.

The references are not as per the journal’s format.

       References have been revised as required.

Thanks again for all you do.

Round 3

Reviewer 1 Report

Please check if anything is missing in Figures 36 and 37.

English language seems to be fine